# 150 MHz polymer resonator for optoacoustic mesoscopy based on a tapered optical fiber

Okan Ülgen [1,2], Tai Anh La [1,2], Christian Zakian[1,2], Maximilian Gotsch [1,2] & Vasilis Ntziachristos [1,2,3] ✉

Optical ultrasound detection enables greater miniaturization than conventional piezoelectric transducers while preserving high sensitivity. Although sub-micron silicon photonics detectors have been demonstrated, image artifacts caused by surface acoustic waves interference remain a key challenge. Polymer detectors offer better acoustic coupling, yet they have been limited to tens of micrometers in size because of optical confinement requirements. Here we overcome that limit with the smallest polymer resonator built on an optical fiber, using a 6 μm thick polymer cavity on a tapered single mode fiber tip. The detector achieved a bandwidth of about 150 MHz and a noise equivalent pressure density of about 1.5 mPa.Hz$^{-1/2}$. Imaging experiments yielded 7 μm axial and 17 μm lateral resolution, with high fidelity performance that surpassed piezoelectric and state of the art optical detectors. This combination of broad bandwidth, artifact free imaging, and manufacturability makes the detector ideal for optoacoustic mesoscopy (OptAM) applications.

Optoacoustic mesoscopy (OptAM) offers an optimal operational specification for an optoacoustic modality, since it allows penetration deeper than optical or optoacoustic microscopy using focused light[1,2]. High fidelity imaging performance is based on broadband detection of ultrasound waves that are excited by short light pulses, the latter acting as delta function excitation stimulating wideband pressure resonances within the object imaged. Systems capturing broadband responses spanning from a few MHz to over 100 MHz have demonstrated superior imaging performance compared to those acquiring bandwidths limited to only a few tens of MHz[2]. This is exemplified by Raster-Scan Optoacoustic Mesoscopy (RSOM), where a high bandwidth enables higher-resolution imaging at shallower depths, facilitating the visualization of fine structures, such as skin vasculature with diameters as small as 7–25 μm[2,3]. Structures extending from the skin surface to the epidermal-dermal junction possess substantial frequency components above 100 MHz, underscoring the necessity for high bandwidth detectors for visualization of fine structures[3,4]. However, implementations of the technique in miniaturized designs are challenging when using piezo-electric (PZT) ultrasound detectors,

both in terms of the bandwidth offered and the quadratic loss of sensitivity with size[5], and may lead to reduced acceptance angles and the detection bandwidths achieved. The drive for miniaturization of ultrasound detection for different applications whereby size becomes critical, such as micro-endoscopy[6,7] or development of hybrid optical–optoacoustic microscopes[8], has led to methods that use optical sensing of ultrasound[5]. Optical detection of ultrasound, typically based on optical interferometry, offers sensitivity that does not depend on the size of the detector area[9–13]. Nevertheless, despite the sensitivity advantage, not all optical designs offer broad detection bandwidths or optimal ultrasound coupling for optoacoustic applications.

Micro-ring resonators of 20 μm diameter achieved noise-equivalent pressure density (NEPD) values as low as 1.3 mPa Hz$^{-1/2}$, but only at bandwidths of 27 MHz, based on the resonant behavior of acoustic membranes[14]. Conversely, sub-micrometer silicon waveguide etalon detectors (SWEDs) demonstrated unprecedented miniaturization and bandwidths of up to 230 MHz[15]. Despite the superior optical properties of detectors built on silicon-on-insulator, the spatio-

[1]Chair of Biological Imaging, Central Institute for Translational Cancer Research (TranslaTUM), School of Medicine and Health & School of Computation, Information and Technology, Technical University of Munich, Munich, Germany. [2]Institute of Biological and Medical Imaging, Bioengineering Center, Helmholtz Zentrum München, Neuherberg, Germany. [3]Munich Institute of Biomedical Engineering (MIBE), Technical University of Munich, Garching b. München, Germany. ✉e-mail: bioimaging.translatum@tum.de

temporal performance was hindered by surface acoustic waves (SAWs), resulting in image distortion and blurring[16]. To reduce SAW interference, polydimethylsiloxane (PDMS) coatings were applied onto a silicon line-detector made of a π-phase-shifted Bragg grating (π-BG)[17]. Asymmetry in the detection field in this case led to a detector with asymmetric lateral resolution, i.e., 5.2 μm along $x$ versus 30.5 μm along $y$. Similar to silicon, silica has a high Young's modulus ($\approx$75 GPa), and thus reduced strain per unit pressure. Strong acoustic reflection (65%) occurs at the silica-water interface because of the suboptimal acoustic coupling caused by the large difference between specific acoustic impedances of silica and water ($Z_s \approx 13.1 \times 10^6$ kg/m²s, $Z_w \approx 1.48 \times 10^6$ kg/m²s). Acoustic impedance mismatch also causes marked SAW generation at the detector surface. As a result, the acoustic resolution of fiber Bragg gratings (FBGs)[18] and π-FBG etalons[19,20] embedded in optical fibers exceeded 45 μm. Strong SAW interference at the silica-water interface is the main limiting factor of detector performance, leading to a larger effective element size and blurring the resulting image.

For an ideal Fabry-Perot interferometer (FPI) detector, high optical finesse and linear response should be paired with the detector's ultrasound matching properties. Polymer Fabry-Perot (FP) detectors show omnidirectional and artifact-free acoustic response due to the improvement in acoustic coupling properties over silicon and silica detectors. Such resonators have been implemented on optical fibers with planar[21,22] or plano-concave[23–25] geometries. However, in contrast to optical detectors that form resonators within optical waveguides, divergent beam walk-off limits the optical phase sensitivity of polymeric resonators. In the case of planar cavities, the non-uniformities in the deposition process are the primary limitation to miniaturizing FPI resonators. Conversely, plano-concave-shaped cavities require a precisely manufactured radius of curvature for the second cavity mirror to reduce beam divergence. Decreasing the cavity thickness alone leads to an increased radius of curvature, thereby increasing optical losses and reducing optical back-coupling efficiency into the fiber core. This trade-off was demonstrated on plano-concave micro-resonators on glass substrates for cavity lengths down to 30 μm[24], whereby a reduction in cavity size from 250 μm to 30 μm increased the bandwidth but reduced the sensitivity. Conversely, when transferring this premise to the tip of optical fibers, the sensitivity achieved for a 16 μm cavity increased over the 30 μm plano-concave cavity, whereby the bandwidth decreased, indicating that the performance observed in plano-concave resonators is not transferable to fiber-tip resonators. This limited bandwidth and the omission of SAW suppression led to reduced lateral and axial resolution. Moreover, attention should be paid that the wavelength of the interrogating source can excite one of the cavity modes and it does not fall within the resultant free spectral range, i.e., the spacing between the wavelengths of successive cavity modes. For these reasons, the miniaturization of polymeric resonator-based ultrasound detectors has been restricted to tens of micrometers due to optical confinement concerns. Likewise, the typical detection bandwidths of such detectors are usually below 50 MHz (>20 μm) (Table 1). Attempts have been made to flatten the frequency response by minimizing acoustic diffraction by rounding the fiber tip[26]. However, optical confinement was not optimized, and the detector's aperture was identical to the fiber diameter, making the detector less sensitive and more susceptible to SAW interference. Overall, as is evident, the design of an optical ultrasound sensor necessitates a careful balance among bandwidth, lateral resolution, sensitivity, and SAW interference.

The new design presented herein introduces a paradigm shift by eliminating the need to offer a trade-off of the operational parameters in the quest for high-bandwidth detection for OptAM. Based on data from tests we have performed on many different designs and approaches over the past decade, we hypothesized that a detector design incorporating a polymer resonator (PR) on the flattened tip of a tapered optical fiber (TOF) will optimize the dimensions and coupling efficiency, maximize bandwidth, lateral resolution, and sensitivity, while also reducing SAW interference. This design, which we named TOF-PR, facilitates cavity encapsulation using a mirror with a small radius of curvature and a smaller overall lateral size. This feature is crucial for achieving high optical back-coupling efficiency and reducing lateral beam walk-off. To experimentally validate our hypothesis, we built detectors on tapered fibers with different base diameters and cavity lengths to test the impact of miniaturization. We show that a PR with a base diameter of 24 μm and a cavity length of 6 μm enables an unprecedented, ultra-broadband frequency response of ~150 MHz at −6 dB, and a class-leading minimum detectable acoustic pressure of 1.5 mPa Hz$^{-1/2}$ owing to high cavity finesse and high elasto-optic polymer response, while the small effective detection cross-section (aperture size) reduces lateral blurring and interference from SAWs. The superior sensitivity and resolution of the TOF-PR are evident through direct comparison with state-of-the-art ultrasensitive extrinsic polymer resonators[23–25], as well as with an intrinsic SWED[15], as summarized in Table 1. Although extrinsic polymer detectors have comparable footprints to the TOF-PR, their substantially lower resolution and sensitivity limit their utility for high-resolution imaging in challenging environments, such as endoscopic applications. Conversely, SWED offers a considerably higher bandwidth, but TOF-PR can achieve comparable axial and lateral resolutions by reducing SAW interference. Moreover, TOF-PR fabrication only requires a wet lab environment, which reduces manufacturing time and cost by approximately 100-fold. We experimentally characterize the merits of the new detector in the laboratory and demonstrate its feasibility for use in optoacoustic imaging applications in vivo. We further showcase the adaptability of the innovative design by incorporating it into a multi-core optical fiber probe capable of simultaneous parallel excitation and detection of optoacoustic waves and discuss the critical manufacturability advances of the TOF-PR detector over the most advanced non-fiber-based silicon photonics detectors like the optical micro-machined ultrasound sensor (OMUS)[14], SWED[15] or silicon-photonics acoustic detector (SPADE)[17].

In this work, we introduce a tapered optical fiber polymer resonator ultrasound detector that overcomes long-standing trade-offs between bandwidth, sensitivity, resolution, and surface acoustic wave interference in optoacoustic mesoscopy. Sensitivity, frequency response, and spatial resolution are experimentally characterized, showing that bandwidth enhancement brings the TOF-PR into the same operational regime as silicon photonics detectors while reducing SAW-induced artifacts. The detector's performance is finally validated through in vivo optoacoustic imaging of the mouse ear, demonstrating its suitability for high-resolution optoacoustic imaging.

## Results

The proposed plano-concave polymer cavity design comprises silver mirrors built at the tip of a single-mode, polarization-maintaining (PM), flat-cone-shaped optical fiber (Fig. 1). The hypothesis was that, unlike using a flat tip, polishing a fiber tip into a cone shape and thereby reducing the fiber tip diameter is a particular element that could impart high lateral resolution and SAW minimization. To achieve a cone-shaped tip, we cleaved (Fig. 1a panel I) and then polished the fiber tip at an angle, and then further polished the fiber end-face, to reach a flat-cone shape (Fig. 1a panel II). The dimensions of the cone and its plateau were adjusted according to the polishing specifications (Supplementary Fig. 1). The flat plateau serves as the base for a PR (Fig. 1a panel III). The reduced base diameter of the resonator ensures a small effective detection area, reducing SAWs that could degrade lateral resolution. Our design and manufacturing methods allowed narrowing of the base diameter to match the mode field diameter (10.1 μm) or even the core diameter of the optical fiber (8.5 μm). This adjustment, however, led to a thinner cavity, resulting in a free spectral range that

**Table 1 | Comparison of our detector (TOF-PR) with other optical fiber-based polymer resonators and SWEDs in the literature by dimension, bandwidth, and sensitivity**

| | Dimensions | | Bandwidth | | Sensitivity | | Resolution | |
|---|---|---|---|---|---|---|---|---|
| | Footprint | Thickness [µm] | −3 dB [MHz] | −6 dB [MHz] | NEPD [mPa Hz$^{-1/2}$] | NEPD x Area [mPa mm² Hz$^{-1/2}$] | Lateral [µm] | Axial [µm] |
| TOF-PR | Ø 24 µm | 6 | 110 | 149 | 1.5 (25 MHz) | $9.0 \times 10^{-4}$ | 17 | 7 |
| 23 | Ø 125 µm | 21 | 25 | 30 | 11 (25 MHz) | $1.3 \times 10^{-1}$ | >38[a] | >35[a] |
| 24[b] | Ø 125 µm | 16 | 20 | 25 | 2.1 (20 MHz) | $2.5 \times 10^{-2}$ | >47[a] | >45[a] |
| 25 | Ø 125 µm | 20 | – | 30 | 40 (30 MHz) | $4.9 \times 10^{-1}$ | 84 | 231 |
| 22 | Ø 125 µm | 30 | 25 | – | 70 (25 MHz) | $4.3 \times 10^{2}$ | >37[a] | >36[a] |
| 15 | 3 mm×0.8 mm | 9 | – | 230 | 9 (25 MHz) | $9.9 \times 10^{-7}$ | 16 | 5 |

In the *NEPD* column, the frequency bandwidth used for characterizing *NEPD* is indicated in brackets.

*NEPD* noise-equivalent pressure density, *TOF-PR* tapered optical fiber-polymer resonator.

[a]Estimates for lateral and axial acoustic resolution are derived from the corresponding read-out beam size of the detector and its high cut-off frequency. In practice, the maximum achievable resolution will exceed the estimated values, primarily due to the coupling of surface acoustic waves to the effective detection region, which results in a detector aperture larger than the read-out beam size.

[b]Only for the 16 µm detector all key parameters are available and reported in the table; for the 12 µm detector only a bandwidth of 55 MHz (presumably −6 dB) is given.

extended beyond the operational range of our interrogation source. To address this, we finely tuned the base diameter and cavity length to secure optical resonance within the 1520–1630 nm wavelength range.

The principle of operation, which is based on Fabry-Perot Interferometry, is shown in Fig. 1b. Under resonance conditions, energy accumulation within the optical resonator led to a significant reduction in the reflected light intensity. For ultrasound detection, a tunable sweep source with a narrow spectral linewidth (<0.0124 pm) is employed (Fig. 1b-I). The optical interrogation wavelength was tuned to the resonance slopes or inflection points (i.e., quadrature points) where the resonance condition varied. This process is referred to as "off-resonance tuning" (Fig. 1b-II). Exposure of the polymer optical resonator to ultrasound induced physical displacements of the mirrors and elasto-optic effects in the medium. The impact of physical displacements was more pronounced than that of elasto-optic effects, allowing the latter to be neglected in the ultrasound detection analysis. The physical displacements cause amplitude modulation of the reflected light intensity, which is the primary operational mechanism for the optical resonator-based ultrasound detector (Fig. 1b-III).

To demonstrate the impact of miniaturization on acoustic detection response, we constructed three PRs with varying cavity lengths (35 µm, 10 µm, 6 µm) and corresponding base diameters (125 µm, 40 µm, 24 µm), designated as PR-I, PR-II, and PR-III, respectively (Fig. 1c). The frequency response of each detector demonstrated an inverse correlation between cavity length and detection bandwidth. Consequently, the −6 dB detection bandwidths were measured using an ultra-wideband optoacoustic point source generated on a 200-nm-thick gold layer (Supplementary Fig. 2c). We found the bandwidth to be ~23 MHz for PR-I, ~100 MHz for PR-II, and ~150 MHz for PR-III. Due to its larger size, PR-I exhibited multiple local minima and maxima in its frequency response, with a −6 dB cut-off frequency of 50.1 MHz, excluding an intensity drop at 27.2 MHz. In contrast, PR-II displayed a smoother frequency response, which declined sharply after 90 MHz. The smallest resonator, PR-III, exhibited optimal performance with a smooth frequency response, indicating the effectiveness of TOF-PR design. PR-III had a −6 dB cut-off frequency at 165 MHz and achieved a bandwidth of approximately 110 MHz at −3 dB and 150 MHz at −6 dB (Fig. 2a). The cavity thickness also inversely correlated with the optical frequency interval between resonance peaks, known as the free spectral range of the resonator. To ensure a resonance peak within the spectral range of our interrogation laser (1520–1630 nm), we employed the described dip-coating method with epoxy to adjust the cavity thickness. Resonances within the specified spectral range were confirmed at cavity thicknesses of 35 µm, 10 µm, and 6 µm.

The detection of longitudinal waves versus the interference generated from SAWs was inferred on B-scans over a broadband ultrasound point source for each detector. Collected sonograms (Fig. 2b) show optoacoustic signals from an ultra-broadband point source captured using detectors with different dimensions. The acquired signals corroborated that miniaturization of the effective element size was the means for high-fidelity detection of longitudinal waves with reduced interference from SAWs. The relative intensity of SAWs was stronger with the standard plano-concave resonator constructed at the tip of a 125 µm-diameter optical fiber, whereas the detector with the smallest detection cross-section (24 µm) primarily detected direct-impact longitudinal waves as the S-wave signal amplitudes were relatively smaller. SAWs are excited at the solid-liquid interface when longitudinal waves (L-waves) are incident at angles near the Rayleigh angle[27]. In sensors with a larger base diameter, the total sensor area exceeds the ultrasound-sensitive area, increasing susceptibility to SAW excitation. By reducing the size of the polymer resonator to closely match the effective sensitive area, the impact of SAWs is minimized. The taper is therefore essential to eliminate the unused part of the optical fiber, which would otherwise act as a source of SAWs[15]. Due to the orientation of the taper, the Rayleigh criterion is not met at the polymer-water interface in this area, effectively suppressing SAW generation. Furthermore, the taper does not interfere with beam propagation, as the minimum taper diameter is more than twice the mode field diameter (MFD) of the fiber. Consequently, no relevant amount of light is carried within the tapered area of the fiber. Moreover, with the miniaturization of the resonator, the reflection of waves and ringing effects were reduced to a minimum. We characterized the directional responsivity of the detectors using B-scan sonograms filtered at bandwidths of 2–75 MHz and 2–150 MHz. The full acceptance angles (at −6 dB) for detectors with base diameters of 125 µm, 40 µm, and 24 µm were measured to be 80°, 104°, and 127°, respectively. The interference of guided waves caused a strong minimum in the responsivity of the 125 µm-resonator at an incidence angle of 22°, whereas this phenomenon was subtle for the miniaturized 24 µm-resonator. Downsizing the effective element size improved the isotropy of the response, even though frequency-dependent directivity effects were unavoidable due to stronger reflection and attenuation losses of acoustic waves at higher frequencies. In accordance with the theory and directional response models for interferometric ultrasound detection, these findings validated our approach for achieving enhanced acoustic response using miniaturized detectors with reduced effective element size[28].

We further characterized the spatial impulse response (SIR) of the PRs (Fig. 2c), with line scans over an ultra-wideband optoacoustic point source. Point spread function (PSF) projections along the *yz*-plane

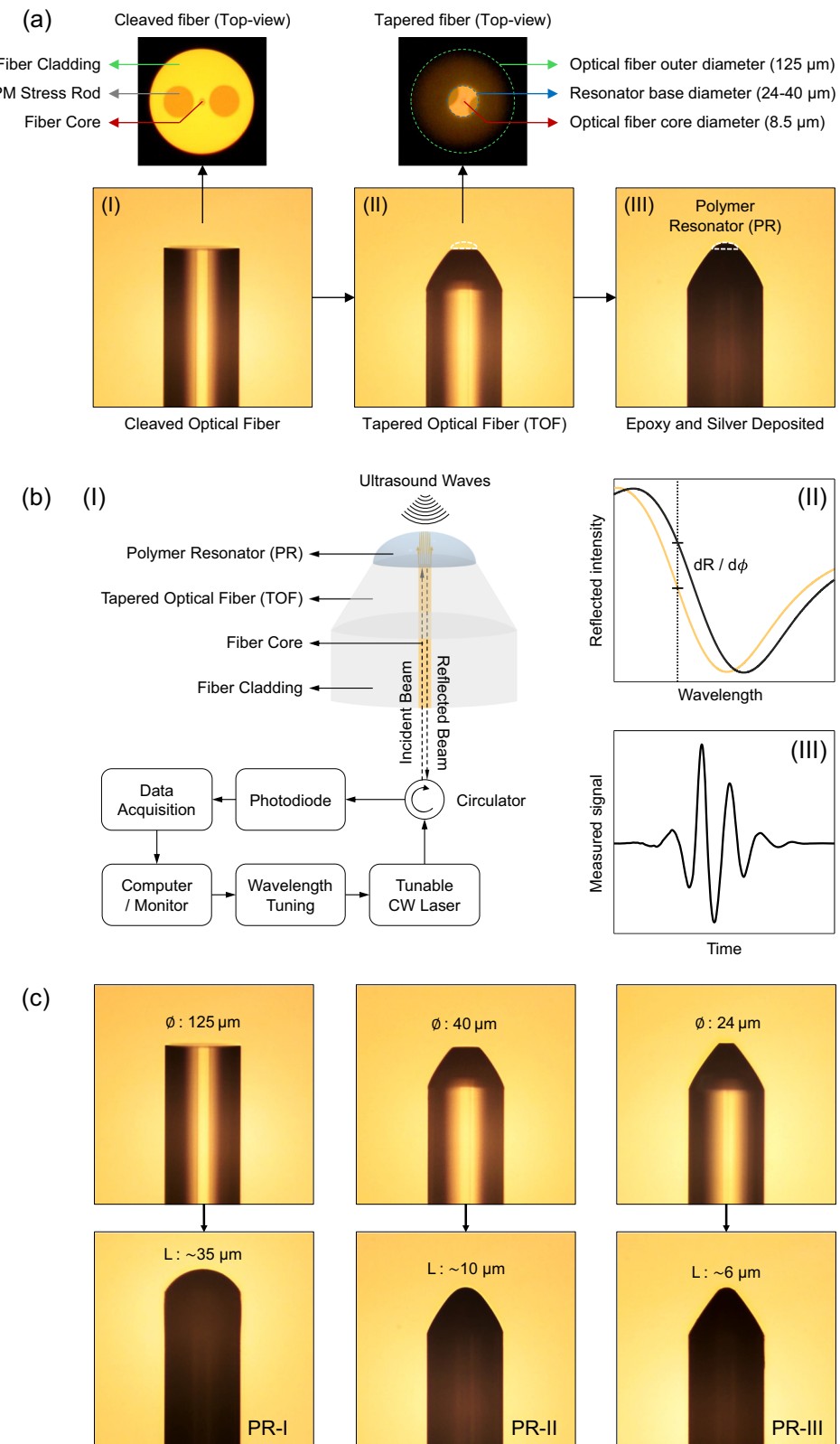

were reconstructed using a 3D inversion technique in the frequency domain previously developed by our research group[29]. The results show that detector miniaturization improved the SIR along axial and lateral directions. Additionally, PSF projections obtained with the 125 μm detector showed diagonal image artifacts around the object. These artifacts, which distorted the reconstructed shape of the object, were caused by SAW interference and non-injective mapping of the

point source in tomographic reconstruction. The miniaturization of the base diameter down to 24 μm, in contrast, eliminated artifacts and distortions. Consequently, the PR-III detector (24 μm base diameter) achieved the highest resolution of 17 μm in the lateral direction and 7 μm in the axial direction, as denoted by the full-width-half-maximum (FWHM) of the PSF at a 240-μm distance (Fig. 2d). Intensity profiles over the reconstructed image of optoacoustic point source (Fig. 2d)

**Fig. 1 | Detector design, working principle, and prototype dimensions.**
**a** Tapered optical fiber polymer resonator (TOF-PR) design. **a-I** Flat-cleaved single-mode optical fiber. The inset shows the cross-section (top-view) of the cleaved optical fiber with a core diameter of 8.5 μm and polarization-maintaining (PM) stress rods within a cladding diameter of 125 μm. **a-II** Tapered optical fiber after polishing into a flat-cone shape. For angled polishing, the fiber was positioned at an angle of approximately 30 degrees on a custom-made polishing disc and rotated along its axial axis (optical axis) during the polishing procedure. The top view shows the resulting reduced base diameter concentric around the fiber core. **a-III** Fiber-based micro resonator after polymer deposition, UV curing, and silver deposition. The fiber appears dark after silver deposition, due to the lower transmittance through the coating. **b** Working principle of the TOF-PR detector. **b-I** A tunable continuous-wave (CW) laser, coupled with a single-mode fiber, is used for optical interrogation. A fiber-based circulator routes the reflected light to a photodiode. A digital oscilloscope is used for simultaneous monitoring of the reflected signal intensity while the signals are being recorded to a computer through a data acquisition (DAQ) board. **b-II** The wavelength of the interrogation source is set to the off-resonance slope (quadrature point), which provides maximum optical phase sensitivity. **b-III** The ultrasound pressure impeding on the polymer resonator causes physical displacements of the mirrors, leading to amplitude modulation in the reflected light intensity. **c** Optical fiber-based polymer resonators (PRs) with various dimensions. PRs with base diameters ($\varnothing$) of 125 μm (PR-I), 40 μm (PR-II), 24 μm (PR-III) were constructed with the polishing and dip-coating processes. The measured cavity lengths (L) are 35 μm for PR-I, 10 μm for PR-II, and 6 μm for PR-III. Adapted from images by the author (O. Ülgen) originally published in the thesis "Miniaturized Optical Ultrasound Detectors for High-Resolution Optoacoustic Imaging", Technical University of Munich, 2023[41].

show that the resolution of PR-I (125 μm base diameter) is limited to 35 μm in the lateral direction and 23 μm in the axial direction. PR-II (40 μm base diameter) has a lateral resolution of 23 μm and an axial resolution of 13 μm. These values are very similar to the analytical estimation of acoustic spatial resolution for finite-size flat aperture ultrasound detectors[30]. The lateral extension of the PSF was proportional to the base diameter of the PR, which ultimately determines the detector aperture size. Additionally, we observed that the SAWs from a larger area were coupled into the optical interrogation region, increasing the effective element size of PR-I in comparison to initial read-out beam radius (Fig. 2b). Therefore, the spatial resolution in the resulting reconstructed image was further diminished, particularly in the lateral direction. Reducing the base diameter to 24 μm effectively minimized SAWs and their detrimental impact, thereby enhancing image resolution and fidelity. Figure 2e, f shows the impact of miniaturization on acoustic spatial characteristics of the PSF. The axial resolution is influenced by the increased cut-off frequency, which results in a narrower PSF as cavity sizes are reduced[15]. In addition, lateral resolution improves with a smaller detector aperture and higher axial resolution[15]. As the OptA signal detected corresponds to the surface integral over the detector's sensitive area, reducing the detector aperture decreases averaging of the acoustic field. Hence, larger apertures alter the spatial frequency components by functioning as a low pass filter, thereby broadening the PSF and reducing resolution[30,31]. In addition, the presence of SAW can further broaden the PSF and introduce image artifacts. The lateral-FWHM and axial-FWHM of the PSF with respect to imaging depth, measured over a range of 240 μm to 4080 μm, are presented for each detector. Due to limited-view scanning and frequency-dependent ultrasound attenuation, the acoustic field strength inversely correlated with imaging depth. Among the characterized detectors, PR-I showed the most rapid degradation in spatial impulse response, particularly in the lateral direction (Fig. 2e), which degraded at twice the rate of the axial direction (Fig. 2f). The blurring in axial direction was primarily caused by the decreasing optoacoustic excitation laser fluence and the stronger attenuation of ultrasound waves with increased imaging depth[32].

To characterize the sensitivity, detector PR-III was selected, and optoacoustic signal detection was performed in a transmission mode setup. A calibrated needle hydrophone (Precision Acoustics, UK) and the fiber detector were placed sequentially 350 μm from the optoacoustic source for signal acquisition. During sensitivity characterization, the optical power of the interrogation source was set to 8.5 dBm. We observed a linear relationship ($R^2 = 0.99$) between the optical resonance slope and the detector sensitivity that was proportional to the optical power of the interrogation laser. The polymer resonator was capable of confining high optical power up to 9.0 mW, a threshold determined by the maximum power of the interrogation laser, while consistently maintaining a linear response throughout operation. Due to the low absorption and coefficient of thermal expansion (CTE) of the adhesive (NOA 68, Norland, USA) within the C-Band, minimal opto-thermal effects were observed. Consequently, cavity heating induced by the interrogation laser was minimized, resulting in high stability during operation.

The PR-III detector demonstrated a detection sensitivity of 126 mV/kPa (Fig. 3a). In the absence of a calibrated ultrasound transducer with a bandwidth comparable to the PR-III, we employed the NEP extrapolation method, as described in ref. 15, which is based on the assumption that a wideband point source on the gold plate generates a flat frequency response, ensuring that ultrasound pressure is uniform across equal bandwidths. We estimated an NEP of 7.3 Pa over a 25 MHz bandwidth, centered around 80 MHz, which corresponds to a NEPD of 1.5 mPa Hz$^{-1/2}$.

The detector produced a linear response highly correlated with the amplitude of generated optoacoustic signals ($R^2 = 0.998$). This linear detector response was verified for optoacoustic excitation pulse energies measured up to 320 nJ, which corresponded to an acoustic pressure of 1.33 kPa at a working distance of 0.5 mm (Fig. 3b). Although higher acoustic pressure values could have been measured, the excitation laser pulse energy was limited to 320 nJ to prevent damage to the thin gold layer sample. Moreover, we investigated the impact of ultrasound attenuation on the detected signal amplitude by adjusting the vertical position of the detector, varying the distance from the point source between 240 μm and 4.76 mm. The peak amplitude of the measured signal decreased at a rate anticipated by theoretical studies and by experiments quantifying the frequency-dependent attenuation of ultrasound waves ($a_0 = 0.00217$ dB MHz$^{-1}$ cm$^{-1}$ in water)[32]. The normalized peak amplitude with respect to distance from the optoacoustic source, measured by the detector over a bandwidth of 25 MHz around its central frequency at 80 MHz, confirmed an exponential decay (Fig. 3c).

To demonstrate the suitability of the detector for imaging applications and validate its performance on complex biological structures, we first imaged, ex vivo, the ear micro-vasculature of an athymic nude mouse. The region of interest, measuring 6 mm × 6 mm, was raster-scanned using motorized stages with a step size of 10 μm. For the raster scan OptAM image reconstruction, we used a back-projection algorithm in the frequency domain[29]. As a result of high sensitivity of the detector (PR-III), an imaging depth of about 1 millimeter at a distance of 480 μm from the tissue could be attained using low fluence. The broad bandwidth achieved with the PR-III detector allowed visualization of fine features (Fig. 4). Its symmetrical response across an ultra-wide acceptance angle (127° at −6 dB bandwidth) enabled accurate imaging of micro-capillaries in various orientations with isotropic lateral resolution characterized to be 20 μm lateral and 7 μm axial resolution. We noted that the bandwidth employed in imaging was broader than the −6dB limit, since sufficient signal to noise ratio was achieved at up to 175 MHz for the PR-III detector (Fig. 2a).

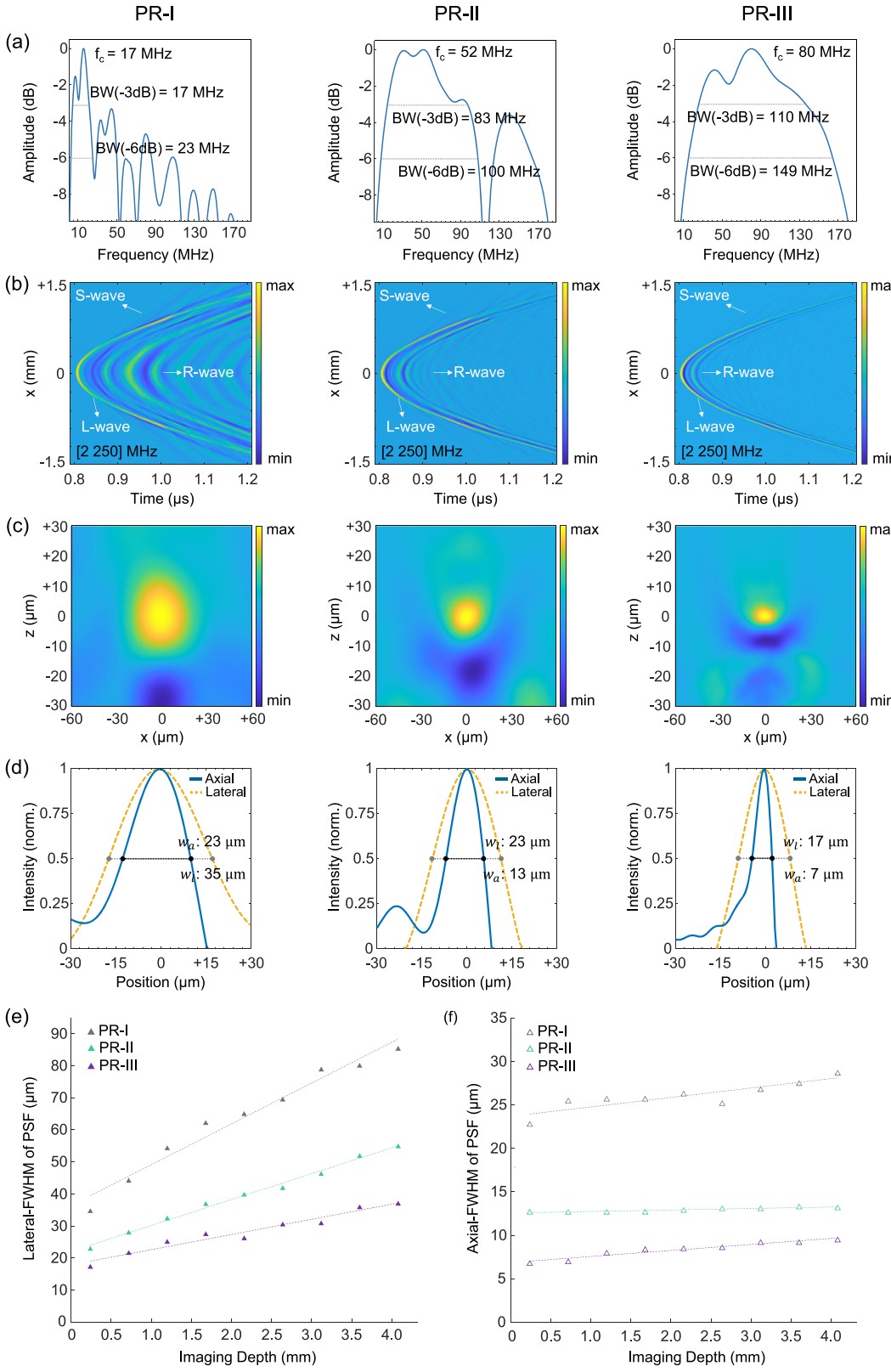

The maximum intensity projection of the reconstructed volumetric data, comprising 5600 slices, renders the 5–50 MHz frequency band in red and the 50–175 MHz frequency band in green (Fig. 4a). This frequency-based color coding helps distinguish vessels by size. The overlay of the two frequency bands shows low-frequency content in orange and high-frequency content in green. Frequency-unmixed images enable precise

differentiation between fine and larger vessels, enhancing the visualization of the dimensional features.

To demonstrate the performance and achievable resolution, we scanned a 2 mm × 2 mm scan area at the mouse ear, with a scan step of 5 μm (Fig. 4b). The captured data was filtered to bandwidth of 5–175 MHz before reconstruction. We highlight two closely positioned vessels in the reconstructed images (Fig. 4b boxes I and II) to

**Fig. 2 | Spatio-temporal response characteristics of the detectors. a** Acoustic spectral responses and detection bandwidths (BW). The central frequency $f_c$ is 17 MHz for PR-I, 52 MHz for PR-II, and 80 MHz for PR-III. The bandwidth at −6 dB is 23 MHz for PR-I, 100 MHz for PR-II, and 149 MHz for PR-III. **b** Spatio-temporal response of the detectors from B-scans over a wideband ultrasound point source generated on a gold plate with a thickness of 200 nm. Longitudinal waves (L-wave), surface acoustic waves (S-wave), and reflected waves (R-wave) are indicated with arrows. Comparing the L-wave, S-wave, and R-waves reveals that the acoustic response of the sensors changes significantly depending on the dimensions of the polymer resonators (PRs). **c** Projections of point-spread-function (PSF) in the XZ-plane acquired at a distance of 240-μm from a wideband point source using the detectors PR-I (23 MHz), PR-II (100 MHz), and PR-III (149 MHz). **d** Axial (blue solid line) and lateral (orange dashed line) line profiles over the imaged point source at a

depth of 240 μm. Axial resolution $w_a$ is measured to be 23 μm for PR-I, 13 μm for PR-II, and 7 μm for PR-III. The lateral resolution $w_l$ is measured to be 35 μm for PR-I, 23 μm for PR-II, 17 μm for PR-III. **e** Lateral full-width half-maximum (FWHM) of PSF with respect to imaging depth, including lines of best fit. The minimum imaging depth is 240 μm, and the maximum imaging depth is 4080 μm. Within this range, the lateral-FWHM of PSF increases from 34.8 μm to 85.5 μm for PR-I, 23 μm to 55 μm for PR-II, and 17.4 μm to 37.2 μm for PR-III. **f** Axial-FWHM of PSF with respect to imaging depth, together with lines of best fit. The impulse response in the axial direction is less dependent on imaging depth compared to the lateral direction. The axial-FWHM of PSF increases from 22.8 μm to 28.7 μm for PR-I, 12.7 μm to 13.2 μm for PR-II, and 6.8 μm to 9.5 μm for PR-III. Adapted from images by the author (O. Ülgen) originally published in the thesis "Miniaturized Optical Ultrasound Detectors for High-Resolution Optoacoustic Imaging", Technical University of Munich, 2023[41].

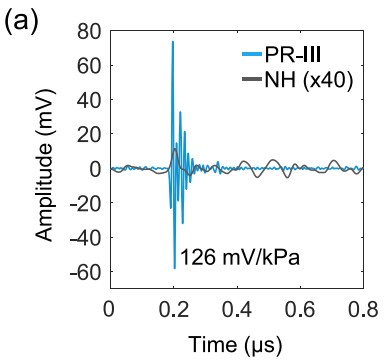
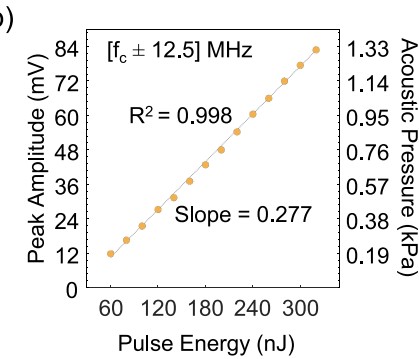
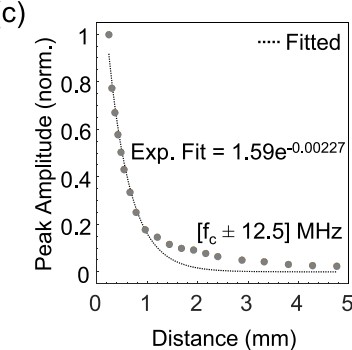

**Fig. 3 | Ultrasound detection using fiber-based polymer resonators. a** Temporal responses recorded with the fiber-based polymer resonator (PR-III) and a calibrated needle hydrophone (NH). The signal acquired with the needle hydrophone was multiplied by a factor of 40 to enhance visibility. The signal acquired with the PR-III was filtered to [fc ± 12.5] MHz, where $f_c$ is the central frequency of detector's bandwidth. On the other hand, the signal acquired with the NH was filtered to 5–30 MHz. Comparing the peak-to-peak amplitudes of the signals, the fiber sensor exhibited an acoustic responsivity approximately 200 times greater than that of the needle hydrophone. **b** Peak amplitude (mV) and acoustic pressure (kPa) with respect to excitation pulse energy (nJ) measured within the frequency range of [fc ± 12.5] MHz. **c** Normalized peak amplitude with respect to distance (mm) from the optoacoustic source. The exponential fit based on the measurements is $1.59e^{-0.00227}$. The optical power of the interrogation source was set to 8.5 dBm during these measurements. Ref reflected, Norm normalized, Exp exponential. Adapted from images by the author (O. Ülgen) originally published in the thesis "Miniaturized Optical Ultrasound Detectors for High-Resolution Optoacoustic Imaging", Technical University of Munich, 2023[41].

demonstrate the capability to image vessels with diameters smaller than 20 micrometers. The intensity profiles reveal vessel diameters of 18.6 μm and 22.7 μm, with a separation of 12 μm between them. The images obtained provide evidence that the detector geometry and material characteristics successfully counteract the influence of SAWs. Consequently, the optical fiber-based PR generated undistorted and artifact-free images. The volumetric optoacoustic reconstruction of the 2 mm × 2 mm area scanned is shown in Supplementary Fig. 3. In vivo images of the back skin of a mouse were also obtained (see Supplementary Fig. 4), confirming that owing to its high sensitivity, the detector exhibited efficient in vivo imaging ability at a modest fluence of around 2 mJ/cm², which is below the safety thresholds for in vivo procedures.

As an outlook, we incorporated the TOF-PR design into multi-core fiber arrangements to demonstrate the feasibility of combining ultrasound detection and optoacoustic excitation channels within a single optical fiber (Supplementary Fig. 5). This approach offers a promising and more compact alternative design. We showed that the conical tip of the fiber functioned as an axicon lens, creating a Bessel-like beam. The resulting focal spot has the potential to enable optical-resolution optoacoustic microscopy at sub-millimeter distances, while also performing acoustic-resolution imaging in OptAM at greater depths using diffused light. For a scientifically rigorous in vivo demonstration, a beam coupling arrangement to increase excitation power is necessary. However, the design and characterization of such a system are beyond the scope of this study. We anticipate that this configuration will

increase SNR and further drive the miniaturization of optoacoustic imaging systems for space-constrained applications like endoscopy[33]. This expectation is supported by a study by Park et al., in which a tapered fiber was used to deliver focused light, achieving high-resolution imaging without the need for additional optics[34].

## Discussion

In contrast to macroscopic optoacoustic applications performed using transducers operating at a few MHz, OptAM requires broad bandwidths and miniaturized point-like detection[2]. The higher frequency bands of a broad bandwidth are necessary to capture features (such as small vessels and capillaries) at a high resolution, while the lower frequency bands enable the reconstruction of larger objects and interfaces. In other words, the wider the bandwidth, the higher the fidelity of the image reached. In contrast, the higher the frequency, the higher the resolution reached. Even if higher ultrasound frequencies attenuate at a steeper rate in tissue than the lower frequencies, they allow for a more detailed visualization of superficial structures and add to the information in an image. In addition, detection from within spatial "points" offers the better definition of a point detector which is essential in the mathematical inversion applied in techniques, such as raster scan OptAM[35], for accurately forming images.

To date, current fiber-based polymer resonators have been limited to bandwidths of around 30 MHz (Table 1). This limitation affects the achievable resolution in optoacoustic imaging and sensitivity to weaker signals originating from deeper layers. Compared to silicon-

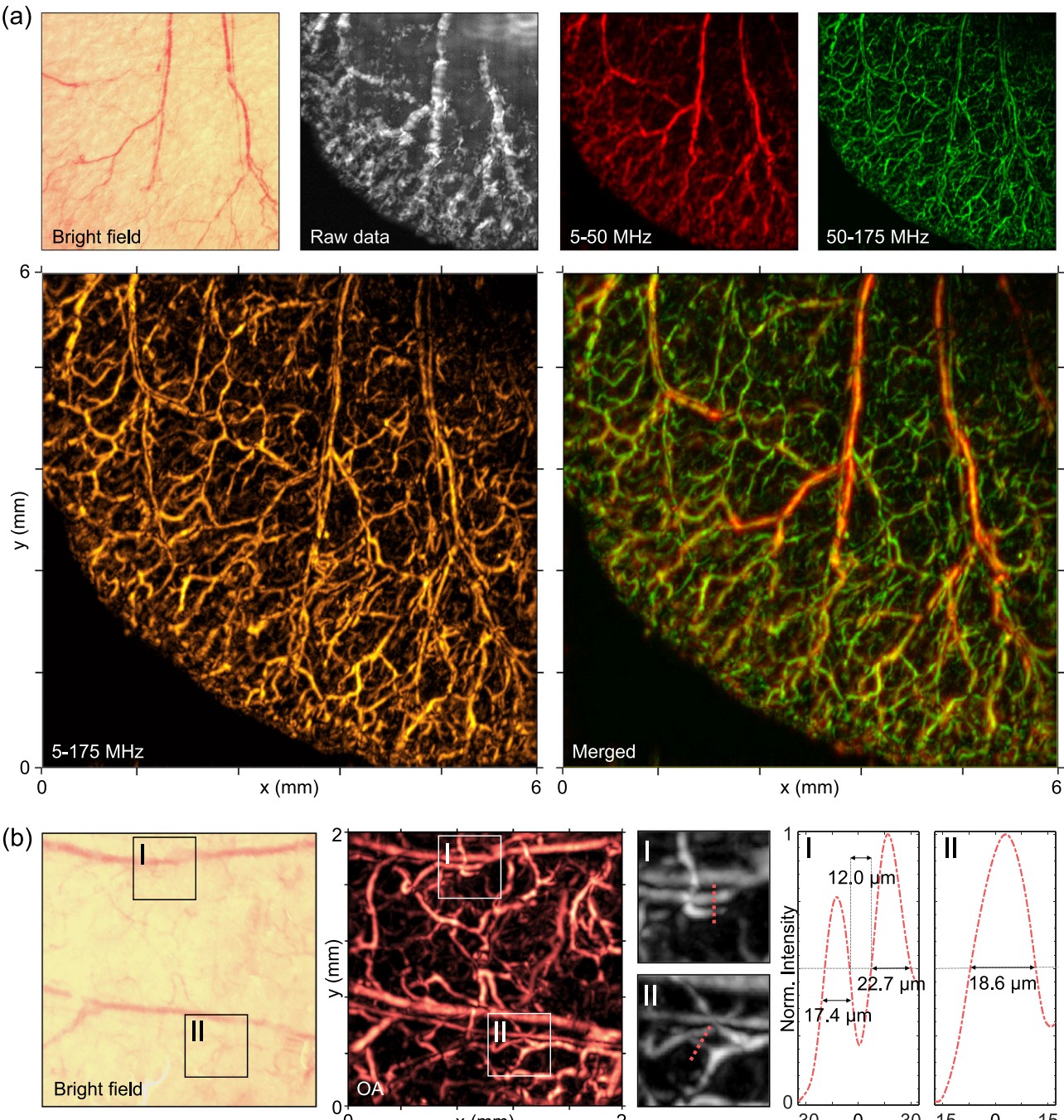

**Fig. 4 | All-optical optoacoustic micro-tomography of a mouse ear ex vivo.**
**a** The microvasculature over an area of 6 mm × 6 mm was raster-scanned with a step size of 10 μm. Top row from left to right: The bright-field image of the same mouse ear serves as a reference for the large vessels; The raw data represents the maximum intensity projection (MIP) of acquired signals prior to reconstruction; The red image shows volumetric data filtered to a bandwidth of [5–50] MHz; The green image shows higher frequency content in the [50–175] MHz range. Bottom row from left to right: MIP based on frequency content between [5–175 MHz] is depicted on the left; The merged image (right) shows the overlap of two frequency bands: [5–50] MHz and [50–175] MHz. High frequency content from smaller vessels appears in green, whereas low frequency content from larger vessels appears in orange. The regions where two frequency bands intersect are represented by yellow color. **b** The framed area on a different mouse ear was raster-scanned ex vivo with a smaller step size of 5 μm to reduce blurring and show finer details of the micro-vasculature. From left to right: The bright field image of the same raster-scanned region was used as a reference; The optoacoustic image shows micro-vessels and capillaries with high contrast; Magnified fields of view (450 μm × 450 μm) I (top image) and II (bottom image); The intensity profiles along the dashed red lines in the magnified images are plotted, and show that the fiber-based detector can resolve microstructures with diameters as small as 17.4 μm and 18.6 μm. Norm: Normalized. Adapted from images by the author (O. Ülgen) originally published in the thesis "Miniaturized Optical Ultrasound Detectors for High-Resolution Optoacoustic Imaging", Technical University of Munich, 2023[41].

based detectors, which operate across a higher frequency range, polymer-based detectors offer advantages, such as lower sensitivity to temperature fluctuations and lower susceptibility to SAW interference due to their small footprint. In order to further miniaturize optical detection of ultrasound, it is essential to develop designs that incorporate materials optimized for acoustic coupling while minimizing susceptibility to SAWs. Decoupling the selection of materials for the fiber and the cavity enables the acoustic and optical properties to be

optimized independently. The design presented in this study successfully combines high sensitivity with uniform 150 MHz broadband ultrasound detection. Until now, this has typically required large PZT transducers, which limit probe miniaturization by blocking optoacoustic illumination. Consequently, our detector enables new space-constrained optoacoustic applications, such as miniaturized sensors or endoscopy. By introducing the TOF-PR design, we demonstrate a new specification for optical detection of ultrasound and a manufacturability that is accessible with simple laboratory equipment versus needing large foundries and expensive infrastructures. The short cavity length (6 μm) and small aperture size (24 μm) of the TOF-PR enabled ultra-broadband (~150 MHz at −6 dB) responses, leading to imaging resolutions of at least 12 μm in the lateral dimension and 7 μm in the axial dimension. This is a metric which is challenging to achieve with piezoelectric transducers and previous optical fiber designs (Table 1).

Optical detectors embedded in fibers and silicon chips, such as FBGs and π-FBG etalons, exhibit Q-factors in the order of $10^5$, resulting in high optical phase sensitivity. By using a polymer external resonator, the resonator compensates for its relatively lower Q-factor compared to SWEDs or π-phase shifted Bragg gratings by using materials with superior acoustic properties, resulting in better acoustic impedance matching and higher acoustic phase sensitivity. In contrast to established polymer-detectors, this developed detector achieved high sensitivity and bandwidth comparable to silicon detectors. Moreover, the adverse effects of SAWs are mitigated through the miniaturization of the detector's effective size and the use of polymeric materials. This advancement enables performance similar to silicon detectors with a substantially smaller footprint, which facilitates high-resolution imaging in space-constrained applications like endoscopy. Although detectors embedded in silicon and silica platforms may also lead to high bandwidths, such detectors have a high tendency to produce images with artifacts due to strong SAW interference. These artifacts arise because of poor acoustic coupling properties of silica and silicon. The SAWs generated at the detector's surface expand the effective detection area beyond the nominal physical dimensions of the waveguide and resonator. As such, polymer FP detectors can be advantageous due to their high elasto-optic coefficient. However, the miniaturization of these detectors has been limited to tens of micrometers due to concerns about optical confinement. Consequently, the resultant bandwidth of less than 50 MHz places a limit on high acoustic resolution.

The imaging experiments presented here confirmed the exceptional optoacoustic imaging capabilities of the detector, showcasing high-fidelity and artifact-free performance even when imaging complex biological structures. The TOF-PR detector is expected to be well-suited for deep-tissue imaging, as its characteristics closely resemble those of an embedded etalon resonator (EER), which has demonstrated effective performance at depths of up to 4 mm in phantoms[35]. Given its comparable lateral and axial resolution in the near field, low NEPD, and uniform frequency response, the TOF-PR is expected to offer similar deep-tissue imaging capabilities while minimizing SAWs. Additionally, the development of a powerful homogeneous wide-field epi-illumination that covers the acoustic acceptance field of the detector is expected to further improve the deep-tissue imaging capabilities. In addition, in vivo optoacoustic imaging experiments confirmed that the detector meets the performance requirements for acquiring high-fidelity images at fluence levels well below biological safety limits. In contrast, optical detectors incorporating embedded resonators are highly vulnerable to SAWs, significantly compromising the accuracy of the reconstructed images[36].

To increase bandwidth and improve axial resolution, the cavity thickness could be further reduced below 6 μm. This could be achieved without compromising optical confinement, but it would result in a free spectral range larger than the operational range of the interrogation source (130 nm). Nevertheless, the three different designs developed have showcased in general the adaptability of the TOF-PR design in terms of bandwidth and dimensions and its ability to relax fine specifications when lower bandwidths are required, for example for imaging at greater depths with lower resolutions[37]. Imaging can be carried out at greater depth by increasing the excitation power within ANSI safety limits and ensuring homogeneous light delivery. Increased sensitivity, reflected by a lower NEPD, is required to detect highly attenuated signals from deep structures. The sensitivity can be further optimized by increasing the optical finesse of the cavity with optimized mirrors and cavity geometries, as well as by improving the pressure phase sensitivity by well-matched acoustic impedances between the water and the polymer resonator and high responsiveness to pressure-induced perturbations of the cavity.

In conclusion, the TOF-PR design introduced herein challenges norms in the field by showing that a miniaturized polymer resonator can exhibit NEPD values down to 1.5 mPa Hz$^{-1/2}$ while achieving broadband frequency responses with frequencies of >170 MHz available to imaging applications. This development could lead to a new standard in optoacoustic imaging applications by exploiting small-form-factor detection, which facilitates alignment with the optical excitation setup. The result is improved SNR and imaging speed in mesoscopic implementations, as well as applications with endoscopic probes and miniaturized biomedical sensors[4,33].

## Methods

### Manufacturing of TOF-PR

The manufacturing process started by cleaving and tapering a single-mode, PM optical fiber (PM1550-XP, Thorlabs). The first mirror of the resonator was formed by dip-coating the fiber tip in a freshly prepared Ag diamine (Tollen's reagent) solution[23]. Following sonication and cleaning to eliminate any residual chemicals, the fiber was dip-coated with UV-curable epoxy. To precisely control the cavity thickness, mechanical contact was repeatedly made between the epoxy-coated fiber tip and the clean surface of a microscope slide, incrementally reducing the epoxy buildup on the fiber tip with each contact. Throughout this procedure, the intensity of the beam reflected from the cavity was continuously monitored, and the procedure was stopped when the desired cavity thickness was achieved. At this stage, even with the absence of the silver coating around the epoxy, a portion of the interrogation beam reflects from the epoxy-air interface. The formation of resonance peaks, observed via oscilloscope, confirmed an accurate radius of curvature of the outer surface, thus ensuring efficient optical back-coupling. The epoxy was then cured using UV light (375 nm). Finally, a second silver mirror was applied around the polymer deposit using the same dip-coating method. Deposition of this second mirror around the epoxy resulted in the formation of strong optical resonance notches. The established fabrication process for the TOF-PR detector features strong potential for automation using CNC processes and robots. In particular, automation will ensure the precise and consistent formation of the epoxy cavity, which is highly dependent on tight control of production parameters to achieve optimal detector performance. During all fabrication steps, the reflection spectrum of the detector can be analyzed to identify faulty detectors. For future in vivo applications, a well-established method that can be applied involves protecting the detector with a thin polymer layer on the outer mirror to guard against mechanical influences and corrosion[38,39].

### Characterization

For sensitivity characterization, we used a calibrated 0.5-mm diameter needle hydrophone (Precision Acoustics, UK) with a sensitivity of 438.5 mV/MPa over the frequency range of 5 to 30 MHz. A pulsed laser (Flare PQ HP GR 2k-500, 1.2 kHz Innolight) generating 1.2-ns pulses at a wavelength of 515 nm was used as the optoacoustic excitation source.

An ultra-broadband optoacoustic point source was generated by focusing the laser beam onto a thin (200 nm) gold plate using a microscope objective (PLN 10x, NA 0.25; Olympus, Germany), achieving a diffraction-limited optical focal spot. The optical power incident on the sample was measured to be 315 μW.

## Imaging experiments

Optoacoustic excitation was achieved using a 1 ns-width pulse with a pulse repetition rate of 1.2 kHz. The excitation wavelength was 515 nm. A multi-mode optical fiber adjacent to the PR-III was used to illuminate a one-millimeter-diameter region on the sample in reflection mode. The optical energy at the fiber output was measured to be 6.7 μJ, corresponding to a fluence value of 1.7 mJ/cm$^2$ on the sample.

Due to the wide detection bandwidth, a reduced scan step was used for the 2 mm × 2 mm scan area, enabling finer spatial sampling and improved resolution in accordance with the Nyquist–Shannon sampling theorem. At each step of the raster scan, the signals were averaged 10 times to reduce noise. The total scan time was 28 min, primarily necessitated by extensive averaging required to compensate for relative intensity noise (RIN) from the excitation laser. Additional constraints included thermal laser instability and spatial beam cleaning with pinholes, which limited the available laser power to relatively low levels. Utilizing a more stable excitation laser that enables an increase in power while remaining within the ANSI safety limit for single pulses of up to 20 mJ/cm$^2$ is expected to reduce the total scan time by a factor of four. Incorporating a galvanometer scanner to guide the excitation beam across the sample could further reduce acquisition times. However, this approach will degrade the SNR at peripheral scan regions due to increased incidence angles. Furthermore, using an excitation laser with a higher pulse repetition rate represents an additional opportunity to further reduce the acquisition time. An array-based approach was not selected for this study due to the requirement for individual tuning of each detector to its quadrature point. Nevertheless, the detector design and functionality would allow such an array configuration. However, implementing this approach would require multiple interrogation sources to tune each element individually, thereby increasing the complexity of the read-out circuit. Another method is the use of almost identical low-finesse elements that can be operated at the same wavelength[21,40].

## Reporting summary

Further information on research design is available in the Nature Portfolio Reporting Summary linked to this article.

# Data availability

Data supporting the findings of this work is available via the Zenodo repository (https://doi.org/10.5281/zenodo.19102221). Due to their large size, the complete raw imaging datasets are not included in the repository but can be obtained from the authors upon request.

# Code availability

The image reconstruction and data analysis code are not publicly available, as they form part of licensed technology. The relevant methods and analysis procedures are described in the manuscript.

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

## Acknowledgements

This project has received funding from the European Union's Horizon 2020 research and innovation programme under grant agreement No. 862811 (RSENSE) to V.N., No. 667933 (MIB) to V.N., C.Z. and O.Ü., and No. 732720 (ESOTRAC) to V.N. and C.Z. We thank Dr. Serene Lee for her attentive reading and improvements of the manuscript.

## Author contributions

O.Ü., C.Z. and V.N. developed the detector concept and designed the study. O.Ü. and T.L. optimized the characterization and imaging setup, including the acquisition code. O.Ü. fabricated the detectors, carried out the characterization and imaging experiments, and analyzed the data. O.Ü., C.Z. and V.N. prepared the manuscript. M.G. made substantial contributions during the review process through manuscript revisions and responses to the reviewers. V.N. secured funding and supervised the project.

## Funding

## Competing interests

V.N. is a founder and equity owner of Maurus OY, sThesis GmbH, Biosense Innovations P.C., Spear UG, and I3 Inc. The remaining authors declare no competing interests.
