## [Transparent Peer Review file · Nature Communications]

150 MHz Polymer Resonator for Optoacoustic Mesoscopy based on a Tapered Optical Fiber

Corresponding Author: Professor Vasilis Ntziachristos

Version 0:

Reviewer comments:

Reviewer #1

(Remarks to the Author)

This manuscript reports on a polymer-based optical sensor designed for broadband detection of acoustic signals, with applications that range from superficial tissue imaging to more general sensing tasks. The authors describe the fabrication process, sensor characteristics, and preliminary imaging experiments. While the concept is interesting from a materials-engineering perspective, several substantive concerns limit the manuscript's contribution and suitability for publication in Nature Communications.

Major Concerns

1. A variety of polymer-based optical and photoacoustic detectors offering similar functionality already exist. The authors have not demonstrated a compelling advance—whether in dynamic range, or a unique application niche—that clearly distinguishes this sensor from previous works. Merely stating that the sensor supports “broadband detection” is not sufficient. It is unclear how this device meaningfully surpasses or complements other polymer- or silicon-based sensors widely reported in the literature.
2. Although the paper mentions potential for deep-tissue imaging, the experiments focus on thin phantom or superficial in vivo measurements. Without validation in thicker samples (or in vivo systems deeper than ~0.5 mm), it is difficult to assess whether the sensor's claimed bandwidth and sensitivity truly hold up under realistic scattering and attenuation conditions.
3. For deep-tissue imaging, light scattering, acoustic attenuation, and noise are significant obstacles. The present data does not experimentally show the sensor's performance at depths beyond superficial layers—undermining claims of deep-tissue applicability.
4. The authors' previous work has reported silicon-based sensors with better performance and smaller form factors. It is unclear what advantages this new polymer design provides—e.g., improved flexibility, biocompatibility, or cost—that would outweigh its apparent downsides. A direct comparison of key performance metrics—bandwidth, sensitivity, spatial resolution, and form factor—between this polymer sensor and the authors' prior silicon-based sensors is needed to justify the polymer approach.
5. The manuscript advertises detection above 100 MHz. Yet, with wide-field illumination, it is rare for biological signals to carry meaningful frequency components in that ultra-high range. Higher bandwidth often compromises sensitivity and complicates system design. The authors should provide evidence of signals (in phantoms or tissues) that genuinely require detection at or above 100 MHz.

Minor Concerns

1. Fig. 2e-f appear to have incorrect or mismatched legends, which confuses interpretation of the data.
2. Fig. 2d may not be universally applicable. The stated spatial resolution depends on frequency content, which changes with sample size and signal characteristics; thus, a single resolution value may not hold for all sample types.
3. The mouse ear is typically <0.5 mm in thickness, which does not serve as a convincing demonstration of deep-tissue imaging. Thicker phantom or ex vivo tissues should be tested to confirm real-world imaging depth capabilities.
4. The reported imaging speed seems low compared to array-based imaging approaches, which can capture 2D or 3D data simultaneously. The benefits of a single-element scanning approach need clearer justification if the method is to be deemed competitive.
5. For claims of deep-tissue use, thicker specimens (or animal models) with multiple millimeters of scattering tissue should be employed. Demonstrating robust signal detection at these depths would strengthen the paper's biomedical relevance.

Reviewer #2

(Remarks to the Author)

In this study, an optoacoustic detector was developed by attaching a polymer resonator (PR) to a tapered optical fiber (TOF). The developed TOF-PR is described as having advantages over conventional detectors in terms of reducing surface acoustic waves (SAWs) and achieving miniaturization. Due to these advantages, it demonstrates a broader bandwidth (150 MHz), higher noise-equivalent pressure density (NEPD, $1.5 \text{ mPa}\cdot\text{Hz}^{-1/2}$), and superior resolution (axial: $7 \mu\text{m}$, lateral: $17 \mu\text{m}$) compared to existing detectors. To verify the performance of the developed detector, in vivo experiments were conducted by imaging the vascular network of a mouse ear, with a scan area of $2 \text{ mm} \times 2 \text{ mm}$ and a maximum depth of $850 \mu\text{m}$. The results showed that the microvasculature's detailed structures were clearly visualized, and small vessels with diameters under $20 \mu\text{m}$ were successfully distinguished.

In summary, attaching the polymer resonator to the tapered tip has proven effective. However, I kindly request that the authors review and address the following minor concerns and questions:

1. The color scheme in Figure 2(e) seems incorrect. Could you please confirm and revise as necessary?
2. As the dimensions of the polymer resonator decrease, the resolution improves significantly. Could you provide a detailed explanation of why this phenomenon occurs?
3. Were there any challenges or issues encountered due to the reduced dimensions of the polymer resonator? For example, did you observe any problems such as heat generation or thermal instability?
4. How long did the scanning process for the $2 \text{ mm} \times 2 \text{ mm}$ area take?
5. The imaging depth seems relatively shallow. Are there any potential methods or modifications you would suggest to increase the imaging depth?
6. I found the research utilizing multi-core fibers to combine excitation beams and readout particularly interesting. This system is expected to enhance imaging depth—have you measured the extent of this improvement? Additionally, it seems that in vivo measurements with the detector were not conducted. Could you clarify the reason for this?
7. The study by Byullee Park et al. (2022), *Shear-Force Photoacoustic Microscopy: Toward Super-Resolution Near-Field Imaging*, explores shear-force photoacoustic microscopy using a tapered fiber, which shares similarities with the TOF-PR detector, particularly in leveraging fiber-based architectures for high-resolution optoacoustic imaging. Additionally, the review by Xiaoyi Bao (2023), *Prospects on Ultrasound Measurement Techniques with Optical Fibers*, provides a broad perspective on fiber-optic ultrasound detection, while Xiaoyi Zhu et al. (2024), *High-Speed Innovations in Photoacoustic Microscopy*, discusses advancements in high-speed PAM systems, focusing on improvements in temporal resolution and real-time biological imaging. Citing these works could enrich the discussion by providing valuable context on how fiber tapering enhances resolution, minimizes acoustic interference, and optimizes imaging speed, allowing for a more comprehensive evaluation of the TOF-PR detector's strengths, limitations, and potential improvements.

Reviewer #3

(Remarks to the Author)

After carefully evaluating the manuscript, I find that while the underlying concept of reducing probe diameter to achieve high bandwidth and mitigate surface acoustic wave (SAW) artifacts is promising, the work presented does not sufficiently demonstrate clear advantages over existing photoacoustic microscopy (PAM) systems. The current findings appear more as a proof-of-concept rather than a robust, fully developed approach ready for a high-impact journal. Below are my specific concerns:

1. Emphasizing an “ultra-small probe” advantage without fully validating its utility

The authors highlight the importance of minimizing the probe diameter and reducing SAW artifacts for high-quality imaging in complex or confined geometries. Yet, the actual experiments—using relatively uncomplicated regions such as mouse ear and dorsal skin—do not fully illustrate how this ultra-small probe would outperform existing PAM systems when targeting truly challenging environments (e.g., curved surfaces, narrow cavities, or hard-to-reach tissue sites).

More representative data should be provided—for instance, imaging the inner ear fold, nail bed, gingiva, or similarly complex geometries. If such testing is not feasible, the Discussion should explicitly clarify how this small probe would outperform established systems in the same application scenarios.

2. Lack of direct quantitative comparisons with existing PAM systems

Despite repeatedly citing the advantages of a miniaturized footprint, high bandwidth, and reduced SAW artifacts, the manuscript does not present a side-by-side comparison with other high-frequency PAM systems or fiber-optic transducers under matching conditions. Many reported techniques exist with bandwidths at or above 100 MHz, making direct comparisons (in terms of resolution, SNR, and acquisition speed) essential to justify claims of superiority.

Conduct or include head-to-head imaging comparisons between the proposed probe and at least one established high-frequency PAM (or fiber-based Fabry-Pérot) system. Measures such as SNR, resolution, field of view, and data acquisition time under identical conditions would clarify the true performance gains.

3. Point-scanning time constraints remain unaddressed

Like most high-resolution acoustic or photoacoustic methods, the proposed system uses raster scanning. The manuscript does not detail frame rates, sampling speeds, or strategies to handle motion artifacts during in vivo imaging. This limits the scope of practical applications, especially for larger fields of view or scenarios requiring real-time insights.

Provide explicit information on scanning speed and potential methods to accelerate data acquisition or mitigate motion artifacts. For instance, an array-based or combinatorial scanning method might alleviate the inherent speed limitations of raster scanning.

4. "Multicore fiber" work restricted to proof-of-concept

While the integrative approach for excitation and detection in a single multicore fiber is intriguing for minimally invasive or endoscopic applications, it is only demonstrated in a rudimentary manner. Critical performance indicators—such as achieving uniform illumination, effective coupling efficiency, and imaging depth in biologically relevant conditions—are either not reported or only superficially addressed.

If the authors consider endoscopy a key selling point, they should substantiate it with more concrete data using realistic phantoms or biological tissues. A deeper discussion of coupling efficiencies and beam uniformity is also necessary to solidify claims of practical endoscopic potential.

5. Gap between theoretical advantages and practical feasibility

Although the polymer resonator design and considerations for reducing SAW artifacts are theoretically commendable, the discussion around reproducibility, mechanical stability, and durability of the device (particularly in vivo or clinical environments) is insufficient. The minimal demonstrations provided make it difficult to gauge readiness for clinical or commercial usage.

For publication in a high-impact journal, more rigorous demonstrations of packaging, mechanical reliability, and performance in complex real-tissue scenarios are strongly recommended. At a minimum, the authors should address how they plan to overcome obstacles in manufacturing, device longevity, and quality assurance.

Overall Summary and Recommendation

In its current form, the manuscript does not convincingly demonstrate a transformative step beyond existing high-frequency PAM approaches. The core claims—namely, that an ultra-small, high-bandwidth probe offers significant benefits for complicated or endoscopic imaging—remain mostly speculative, as the data presented focus on relatively simple, flat tissue sites and limited proof-of-concept fiber integration.

Therefore, based on the present scope and evidence, I believe the manuscript's novelty and completeness are not sufficient for publication in Nature Communications. The authors are encouraged to perform additional, more rigorous experiments—especially in complex geometries or genuine endoscopic scenarios—and to provide robust comparisons with state-of-the-art systems. Addressing the practical engineering, reproducibility, and scanning-speed limitations would also greatly enhance the impact and applicability of the work.

Reviewer #4

(Remarks to the Author)

The manuscript titled "150 MHz Polymer Resonator for Optoacoustic Mesoscopy based on a Tapered Optical Fiber" presents a tapered optical fiber polymer resonator (TOF-PR) with a 6 μm thick polymer resonator, achieving a 150 MHz detection bandwidth and a noise-equivalent pressure density (NEPD) of 1.5 $\text{mPa}\cdot\text{Hz}^{-1/2}$, which shows a broader bandwidth and lower NEPD than previous works [1-5]. The authors claim that the miniaturization of the fiber tip and the subsequent fabrication of a thinner Fabry-Perot (FP) cavity represents a significant advancement in optoacoustic detection. However, upon thorough comparison with existing literature, this work lacks the novelty and scientific depth required for publication in Nature Communications.

Major Concerns and Recommendation for Rejection

Limited Novelty and Incremental Contribution: The concept of enhancing bandwidth by reducing the FP cavity thickness was already disclosed in the 2017 Nature Photonics paper [1]. The current manuscript applies this idea by tapering the fibre tip to fabricate a thinner cavity, claiming this as a novel, more accessible fabrication method. However, tapering the fiber tip is not a new strategy—this approach has been thoroughly explored in Zhang et al. (2015) [6] and subsequent works. Thus, this manuscript essentially replicates previously established strategies, making the improvements in acoustic characteristics and bandwidth expected outcomes rather than novel contributions.

Redundancy with Prior Work: The methods used in this study are redundant with prior works. Zhang et al. (2015) [6] demonstrated the optimization of fiber-optic ultrasound sensors through fiber tip modifications, including tapering and cavity design. The Optics Express publications by Marques et al. (2020) [7], Martin-Sanchez et al. (2022) [8], and Martin-Sanchez et al. (2023) [9] provide comprehensive models and experimental validations of Fabry-Perot etalons illuminated by focused beams, Gaussian beam propagation in FP etalons, and plano-concave microresonator behaviour. These studies cover the theoretical foundations and experimental optimizations that this manuscript overlooks, further highlighting its lack of originality.

Neglect of Optical and Acoustic Mechanisms: The manuscript reports a reduction in surface acoustic wave (SAW) artifacts but provides no rigorous physical analysis or modeling to explain this phenomenon. The authors simply observe SAW suppression without exploring the underlying acoustic mechanisms or optimizing the design parameters. Similarly, the optical interrogation of the tapered fiber significantly affects the optical field distribution in FP Cavity, yet the manuscript lacks an analysis of how tapering impacts wavefront propagation, cavity finesse, or sensitivity.

Absence of Design Optimization: Unlike the referenced works, particularly Zhang et al. (2015) [6], which conducted comprehensive design optimizations, this manuscript does not explore the influence of taper angles, polymer thicknesses, or cavity curvatures on device performance. The lack of parametric studies indicates a superficial approach to the research, falling short of the scientific rigor expected for publication in Nature Communications.

Overall Poor Completion of Study: The study appears to be an incomplete extension of the ideas presented in 2017 [1], with minimal modifications to the fiber tip geometry and no additional theoretical or experimental advancements. The lack of depth in both optical and acoustic analyses reflects an underdeveloped study that does not meet the standards of a high-impact journal.

Conclusion and Recommendation

In conclusion, this manuscript lacks the originality, depth, and scientific rigor required for publication in Nature Communications. The primary contribution—tapering the fiber tip to fabricate a thinner FP cavity—is an incremental improvement that does not justify publication in a high-impact multidisciplinary journal. The absence of theoretical analysis, optimization, and comprehensive comparison with existing literature further weakens the manuscript. Given these limitations, I strongly recommend rejection and suggest submission to a specialized optics journal where this work may be more appropriately categorized as a Technical Note.

References

1. Guggenheim, James A., Jing Li, Thomas J. Allen, Richard J. Colchester, Sacha Noimark, Olumide Ogunlade, Ivan P. Parkin et al. "Ultrasensitive plano-concave optical microresonators for ultrasound sensing." *Nature Photonics* 11, no. 11 (2017): 714-719.
2. Li, Guangyao, Zhendong Guo, and Sung-Liang Chen. "Miniature all-optical probe for large synthetic aperture photoacoustic-ultrasound imaging." *Optics express* 25, no. 21 (2017): 25023-25035.
3. Chen, Bohua, Yuwen Chen, and Cheng Ma. "Photothermally tunable Fabry-Pérot fiber interferometer for photoacoustic mesoscopy." *Biomedical Optics Express* 11, no. 5 (2020): 2607-2618.
4. Yang, Liuyang, Dongchen Xu, Geng Chen, Anqi Wang, Liangye Li, and Qizhen Sun. "Miniaturized fiber optic ultrasound sensor with multiplexing for photoacoustic imaging." *Photoacoustics* 28 (2022): 100421.
5. Lin, Wei-Kuan, Linyu Ni, Xueming Wang, Jay L. Guo, and Guan Xu. "Fabrication of a translational photoacoustic needle sensing probe for interstitial photoacoustic spectral analysis." *Photoacoustics* 31 (2023): 100519.
6. Zhang, Edward Z., and Paul C. Beard. "Characteristics of optimized fibre-optic ultrasound receivers for minimally invasive photoacoustic detection." In *Photons Plus Ultrasound: Imaging and Sensing 2015*, vol. 9323, pp. 151-159. SPIE, 2015.
7. Marques, Dylan M., James A. Guggenheim, Rehman Ansari, Edward Z. Zhang, Paul C. Beard, and Peter RT Munro. "Modelling Fabry-Pérot etalons illuminated by focussed beams." *Optics Express* 28, no. 5 (2020): 7691-7706.
8. Martin-Sanchez, David, Jing Li, Dylan M. Marques, Edward Z. Zhang, Peter RT Munro, Paul C. Beard, and James A. Guggenheim. "ABCD transfer matrix model of Gaussian beam propagation in Fabry-Perot etalons." *Optics Express* 30, no. 26 (2022): 46404-46417.
9. Martin-Sanchez, David, Jing Li, Edward Z. Zhang, Paul C. Beard, and James A. Guggenheim. "ABCD transfer matrix model of Gaussian beam propagation in plano-concave optical microresonators." *Optics express* 31, no. 10 (2023): 16523-16534.

Reviewer #5

(Remarks to the Author)

Version 1:

Reviewer comments:

Reviewer #1

(Remarks to the Author)

The authors have addressed my comments fully. Thanks.

Reviewer #2

(Remarks to the Author)

My comments are well addressed.

Reviewer #3

(Remarks to the Author)

I have reviewed the revised manuscript and the authors' response to my previous critiques. While I acknowledge the effort taken to address some of the points, my fundamental concerns regarding the novelty, transformative potential, and practical validation of this work remain largely unresolved. The core issue is that the manuscript presents a solid engineering

optimization within a known design space, but it fails to demonstrate a conceptual leap or a clear, validated advantage that would constitute a significant advance for the broad readership of Nature Communications.

The authors' responses often justify the absence of key experiments by stating they are "beyond the scope of this study" (Line 410). However, for a journal of this stature, precisely those experiments, such as validation in challenging anatomical environments or a direct, system-level performance comparison, define the scope required to substantiate the high-impact claims being made.

Below are a few of my major persistent concerns.

First, the "Ultra-Small Probe" advantage remains largely speculative and untested. The authors argue that the miniaturized footprint is key for space-constrained applications like endoscopy (e.g., Lines 51, 412). However, the imaging demonstrations remain confined to simple, flat, and accessible surfaces (mouse ear, dorsal skin). The rebuttal concedes that imaging in complex regions is "not possible" with their current trans-illumination system. This is a critical omission. A claim of endoscopic utility is severely weakened without data from a curved, confined, or otherwise challenging geometry that would actually benefit from the small size. The provided data does not prove that this probe enables something that existing, larger probes cannot achieve in these same, simple scenarios.

Second, in my personal opinion the work represents incremental advance, but not a paradigm shift. The authors have more clearly articulated their position that the TOF-PR combines the benefits of silicon detectors (bandwidth) and polymer detectors (acoustic coupling). However, this is presented as an engineering trade-off and optimization, not a fundamental breakthrough. The core concepts using a polymer for better acoustic impedance, reducing cavity size for bandwidth, and tapering a fiber tip are previously established, as rightly noted by other reviewers. The authors' work refines these concepts but does not introduce a new principle or mechanism. The performance improvements, while commendable, appear to be an expected outcome of this refinement process rather than a surprising or disruptive discovery. The manuscript reads as a good, specialist engineering paper, but not one that redefines the field's boundaries.

Third, confusion of in vivo and ex vivo. On Lines 362-363, it says "To demonstrate the performance and resolution achieved in-vivo (Suppl. Fig.4), we scanned a 2 mm x 2 mm scan area at the mouse ear, with a scan step of 5 μm (Fig.4b)." But according to the figure caption of Figure 4, the results are from a mouse ear ex vivo.

In summary, the revisions have improved the manuscript's clarity, but the work is an incremental development; the performance demonstrated is strong, but the claim of transformative potential for endoscopy is still not supported by the necessary experimental validation in relevant, challenging environments. For these reasons, I think the work would be more appropriately suited for a leading, but more specialized, journal in the field of optical or ultrasound sensing.

Reviewer #4

(Remarks to the Author)

The authors have addressed all my previous comments. I have no further comments.

Reviewer #5

(Remarks to the Author)

Version 2:

Reviewer comments:

Reviewer #3

(Remarks to the Author)

The authors have addressed my major concerns towards the former version of submission and I have no more comments at this stage. Thank you.

Reviewer #5

(Remarks to the Author)

REVIEWER COMMENTS

Reviewer #1 (Remarks to the Author):

This manuscript reports on a polymer-based optical sensor designed for broadband detection of acoustic signals, with applications that range from superficial tissue imaging to more general sensing tasks. The authors describe the fabrication process, sensor characteristics, and preliminary imaging experiments. While the concept is interesting from a materials-engineering perspective, several substantive concerns limit the manuscript's contribution and suitability for publication in Nature Communications.

We sincerely thank the reviewer for their constructive comments. We have carefully addressed all the points raised and hope that our revised manuscript will now be deemed suitable for publication in Nature Communications.

Major Concerns

1. A variety of polymer-based optical and photoacoustic detectors offering similar functionality already exist. The authors have not demonstrated a compelling advance—whether in dynamic range, or a unique application niche—that clearly distinguishes this sensor from previous works. Merely stating that the sensor supports “broadband detection” is not sufficient. It is unclear how this device meaningfully surpasses or complements other polymer- or silicon-based sensors widely reported in the literature.

We thank the reviewer for highlighting this important point. In response, we have added a paragraph to the manuscript providing a detailed explanation of why our detector outperforms existing detectors.

“To date, current fiber-based polymer resonators have been limited to bandwidths of around 30 MHz (Table 1). This limitation affects the achievable resolution in optoacoustic imaging and sensitivity to weaker signals originating from deeper layers. Compared to silicon-based detectors, which operate across a higher frequency range, polymer-based detectors offer advantages, such as lower sensitivity to temperature fluctuations and lower susceptibility to SAW interference due to their small footprint. In order to further miniaturize optical detection of ultrasound, it is essential to develop designs that incorporate materials optimized for acoustic coupling while minimizing susceptibility to SAWs. Decoupling the selection of materials for the fiber and the cavity enables the acoustic and optical properties to be optimized independently. The design presented in this study successfully combines the high bandwidth typical of silicon-detectors with the small footprint of fiber-based detectors, thereby bringing the resolution and sensitivity of silicon-detectors to space-constrained applications like endoscopy.”

2. Although the paper mentions potential for deep-tissue imaging, the experiments focus on thin phantom or superficial in vivo measurements. Without validation in thicker samples (or in vivo systems deeper than ~0.5 mm), it is difficult to assess whether the sensor's claimed bandwidth and sensitivity truly hold up under realistic scattering and attenuation conditions.

We thank the reviewer for raising the question regarding the potential for deep-tissue imaging. Given the similarities in properties between our detector and an embedded etalon resonator, we anticipate that our detector will be well suited for in vivo imaging at depths greater than 0.5 mm. Testing our sensor in deep tissue would require a powerful and uniform wide-field illumination that covers the entire acceptance field of the transducer, which was not available at the time. Nonetheless, our promising results suggest that integrating our detector into an imaging system equipped with

powerful wide-field epi-illumination arrangement could enable in vivo deep-tissue imaging. However, the development and characterization of such an illumination system falls outside the scope of this study. Nevertheless, we have added the following section to the manuscript to further elaborate on the strong potential of the detector for deep-tissue imaging.

“The TOF-PR detector is expected to be well-suited for deep-tissue imaging, as its characteristics closely resemble those of an embedded etalon resonator (EER), which has demonstrated effective performance at depths of up to 4 mm in phantoms [41]. Given its comparable lateral and axial resolution in the near field, low NEPD, and uniform frequency response, the TOF-PR is expected to offer similar deep-tissue imaging capabilities while minimizing SAWs. Additionally, the development of a powerful homogeneous wide-field epi-illumination that covers the acoustic acceptance field of the detector is expected to further improve the deep-tissue imaging capabilities.”

3. For deep-tissue imaging, light scattering, acoustic attenuation, and noise are significant obstacles. The present data does not experimentally show the sensor’s performance at depths beyond superficial layers—undermining claims of deep-tissue applicability.

We thank the reviewer for the question regarding the potential for deep-tissue imaging. We have addressed deep-tissue imaging in our response to question 2, where we included an additional paragraph and a relevant reference. To further elaborate on the similar properties between our detector and the EER, we refer to Figures 2e and f, which show the lateral and axial resolution, respectively, at different distances from the source in water, taking acoustic attenuation into account. These results closely align with the EER’s resolutions, which exhibits similar bandwidth and sensitivity (NEPD). Accordingly, the TOF-PR detector presented here is expected to perform comparably in phantoms and tissue up to depths of 4 mm.

4. The authors’ previous work has reported silicon-based sensors with better performance and smaller form factors. It is unclear what advantages this new polymer design provides—e.g., improved flexibility, biocompatibility, or cost—that would outweigh its apparent downsides. A direct comparison of key performance metrics—bandwidth, sensitivity, spatial resolution, and form factor—between this polymer sensor and the authors’ prior silicon-based sensors is needed to justify the polymer approach.

We thank the reviewer for the suggestion to include a comparison with silicon-based sensors from previous work. In response, we have expanded the table to present a comprehensive view of the detectors’ dimensions, bandwidth, sensitivity and resolutions.

	Dimensions		Bandwidth		Sensitivity		Resolution	
	Footprint	Thickness [μm]	-3 dB [MHz]	-6 dB [MHz]	NEPD [$\text{mPa}\cdot\text{Hz}^{-1/2}$]	NEPD x Area [$\text{mPa}\cdot\text{mm}^2\cdot\text{Hz}^{-1/2}$]	Lateral [μm]	Axial [μm]
TOF-PR	\varnothing 24 μm	6	110	149	1.5 (25 MHz)	9.0×10^{-4}	17	7
[23]	\varnothing 125 μm	21	25	30	11 (25 MHz)	1.3×10^{-1}	> 38*	> 35*
[24]	\varnothing 125 μm	16	20	25	2.1 (20 MHz)	2.5×10^{-2}	> 47*	> 45*
[25]	\varnothing 125 μm	20	-	30	40 (30 MHz)	4.9×10^{-1}	84	231
[22]	\varnothing 125 μm	30	25	-	70 (25 MHz)	4.3×10^2	> 37*	>36*
[16]	3000 μm x 800 μm	9	-	230	9 (25 MHz)	9.9×10^{-7}	16	5

As shown in the table above, our TOF-PR offers performance characteristics on par with the SWED, while having a smaller form factor, which is advantageous for operation in space-constrained applications, such as endoscopy. TOF-PR achieves comparable lateral and axial resolutions by reducing SAW interference, which causes artifacts and broadens the SWED's point spread function. A detailed explanation is provided in question 3 from reviewer 4. In comparison to established polymer resonators, our detector achieved a twofold improvement in lateral resolution and a fivefold improvement in axial resolution, all while maintaining a similar footprint. Furthermore, the TOF-PR can be developed and manufactured at a much lower cost — typically well under €1 000 — using standard wet-lab equipment, whereas the SWED requires a full chip development process costing several hundred thousand euros and relying on more specialized resources. We have now included the following summary statement comparing our detector to existing detectors.

“The superior sensitivity and resolution of the TOF-PR are evident through direct comparison with state-of-the-art ultrasensitive extrinsic polymer resonators [25],[26],[27], as well as to an intrinsic SWED [16], as summarized in Table 1. Although extrinsic polymer detectors have comparable footprints to the TOF-PR, their substantially lower resolution and sensitivity limit their utility for high-resolution imaging in challenging environments, such as endoscopic applications. Conversely, SWED offers a considerably higher bandwidth, but TOF-PR can achieve comparable axial and lateral resolutions by reducing SAW interference. Moreover, TOF-PR fabrication only requires a wet lab environment, which reduces manufacturing time and cost by approximately 100-fold.”

5. The manuscript advertises detection above 100 MHz. Yet, with wide-field illumination, it is rare for biological signals to carry meaningful frequency components in that ultra-high range. Higher bandwidth often compromises sensitivity and complicates system design. The authors should provide evidence of signals (in phantoms or tissues) that genuinely require detection at or above 100 MHz.

We thank the reviewer for this insightful question. In response, we have added references that demonstrate the advantages of high frequency detection in optoacoustic imaging.

“Systems capturing broadband responses spanning from a few MHz to over 100 MHz have demonstrated superior imaging performance compared to those acquiring bandwidths limited to only a few tens of MHz [2]. This is exemplified by Raster-Scan Optoacoustic Mesoscopy (RSOM), where a high bandwidth enables higher-resolution imaging at shallower depths, facilitating the visualization of fine structures, such as skin vasculature with diameters as small as 7 – 25 μm [3], [4]. Structures extending from the skin surface to the epidermal-dermal junction possess substantial frequency components above 100 MHz, underscoring the necessity for high bandwidth detectors for visualization of fine structures [3], [5].”

Minor Concerns

1. Fig. 2e-f appear to have incorrect or mismatched legends, which confuses interpretation of the data.

We thank the reviewer for bringing this to our attention. In response, we have updated the figures to ensure that the legends match.

2. Fig. 2d may not be universally applicable. The stated spatial resolution depends on frequency content, which changes with sample size and signal characteristics; thus, a single resolution value may not hold for all sample types.

We acknowledge the importance of considering the dependency between resolution and frequency content. We thank the reviewer for this valuable comment and would like to provide further clarification by elaborating on Figure 2d, e and f.

Figure 2d presents the axial and lateral intensity profiles over the imaged point source for all three detectors at an imaging depth of 240 μm . The optoacoustic point source is defined by an excitation laser with a spot size of 4 μm . Given that the source is point-like and broadband, the intensity profiles of the point spread function are limited primarily by the characteristics of the detectors. Larger structures will produce lower bandwidth signals and are therefore not suitable for demonstrating the detectors' capabilities. Additionally, due to signal attenuation, the bandwidth decreases for features located in deeper layers in tissue, which in turn results in decreasing imaging resolution (see Figure 2e,f).

To enhance clarity and ensure readers have the relevant information regarding imaging depth, we have updated the caption of Figure 2d as follows:

“Axial (blue solid line) and lateral (orange dashed line) line profiles over the imaged point source at a depth of 240 μm . Axial resolution is measured to be 23 μm for PR-I, 13 μm for PR-II, and 7 μm for PR-III. The lateral resolution is measured to be 35 μm for PR-I, 23 μm for PR-II, 17 μm for PR-III.”

3. The mouse ear is typically <0.5 mm in thickness, which does not serve as a convincing demonstration of deep-tissue imaging. Thicker phantom or ex vivo tissues should be tested to confirm real-world imaging depth capabilities.

We thank the reviewer for this question and would like to refer to our responses to Questions 2 and 3, where this concern has been addressed.

4. The reported imaging speed seems low compared to array-based imaging approaches, which can capture 2D or 3D data simultaneously. The benefits of a single-element scanning approach need clearer justification if the method is to be deemed competitive.

We appreciate the reviewer’s interest in imaging speed, which is indeed an important aspect of imaging approaches. However, the primary aim of this study is to demonstrate the performance of the TOF-PR sensor rather than to optimize or achieve fast image acquisition speeds. Specifically, this work primarily focuses on the development of a high BW, small footprint and low-cost fiber-optic ultrasound detector. To demonstrate the potential of this sensor, a raster-scanning single-element setup is sufficient. The raster scan speed was mainly limited by the need for extensive averaging, as well as the speed of the translation stages and the repetition rate of the excitation laser.

Nonetheless, we have added a paragraph that highlights the limitations of our imaging setup and discussed potential avenues for increasing acquisition speed if needed, such as improving the light provision, using a galvanometer scanner to deflect the beam faster across the sample or selecting an excitation laser with a higher repetition rate.

“The total scan time was 28 minutes, primarily necessitated by extensive averaging required to compensate for relative intensity noise (RIN) from the excitation laser. Additional constraints included thermal laser instability and spatial beam cleaning with pinholes, which limited the available laser power to relatively low levels. Utilizing a more stable excitation laser that enables an increase in power while remaining within the ANSI safety limit for single pulses of up to 20 mJ/cm^2 is expected to reduce the total scan time by a factor of four. Incorporating a galvanometer scanner to guide the excitation beam across the sample could further reduce acquisition times. However, this approach will degrade the SNR at peripheral scan regions due to increased incidence angles. Furthermore, using an excitation laser with a higher pulse repetition rate represents an additional opportunity to further reduce the acquisition time.”

Furthermore, we have added a statement regarding the potential use of an array-based approach. While the sensor design in principle supports such a configuration, its implementation would necessitate a more complex read-out circuit. The development of such a system, involving multiple interrogation sources, lies beyond the scope of this work.

“An array-based approach was not selected for this study due to the requirement for individual tuning of each detector to its quadrature point. Nevertheless, the detector design and functionality would allow such an array configuration. However, implementing this approach would require multiple interrogation sources to tune each element individually, thereby increasing the complexity of the read-out circuit. Another method is the use of almost identical low-finesse elements that can be operated at the same wavelength [22], [37].”

5. For claims of deep-tissue use, thicker specimens (or animal models) with multiple millimeters of scattering tissue should be employed. Demonstrating robust signal detection at these depths would strengthen the paper's biomedical relevance.

We thank the reviewer for this question and would like to refer to our response to Questions 2 and 3, where this concern has been addressed.

Reviewer #2 (Remarks to the Author):

In this study, an optoacoustic detector was developed by attaching a polymer resonator (PR) to a tapered optical fiber (TOF). The developed TOF-PR is described as having advantages over conventional detectors in terms of reducing surface acoustic waves (SAWs) and achieving miniaturization. Due to these advantages, it demonstrates a broader bandwidth (150 MHz), higher noise-equivalent pressure density (NEPD, $1.5 \text{ mPa}\cdot\text{Hz}^{-1/2}$), and superior resolution (axial: $7 \mu\text{m}$, lateral: $17 \mu\text{m}$) compared to existing detectors. To verify the performance of the developed detector, in vivo experiments were conducted by imaging the vascular network of a mouse ear, with a scan area of $2 \text{ mm} \times 2 \text{ mm}$ and a maximum depth of $850 \mu\text{m}$. The results showed that the microvasculature's detailed structures were clearly visualized, and small vessels with diameters under $20 \mu\text{m}$ were successfully distinguished.

In summary, attaching the polymer resonator to the tapered tip has proven effective. However, I kindly request that the authors review and address the following minor concerns and questions:

1. The color scheme in Figure 2(e) seems incorrect. Could you please confirm and revise as necessary?

We thank the reviewer for pointing this out. The figures have been updated with matching legends for figures 2e and 2f.

2. As the dimensions of the polymer resonator decrease, the resolution improves significantly. Could you provide a detailed explanation of why this phenomenon occurs?

We thank the reviewer for their question regarding resolution improvement from increased bandwidth via cavity size reduction. We have now included a paragraph to clarify the underlying principle.

“The axial resolution is influenced by the increased cut-off frequency, which results in a narrower PSF as cavity sizes are reduced [16]. In addition, lateral resolution improves with smaller detector aperture and higher axial resolution [16]. As the OptA signal detected corresponds to the surface integral over the detector’s sensitive area, reducing the detector aperture decreases averaging of the acoustic field. Hence, larger apertures alter the spatial frequency components by functioning as a low pass filter, thereby broadening the PSF and reducing resolution [34], [35]. In addition, the presence of SAW can further broaden the PSF and introduce image artifacts.”

3. Were there any challenges or issues encountered due to the reduced dimensions of the polymer resonator? For example, did you observe any problems such as heat generation or thermal instability?

We thank the reviewer for their interest in the challenges encountered due to the significant size reduction. We did not observe issues with heat generation or thermal stability. These findings are now mentioned in the text as follows:

“The polymer resonator was capable of confining high optical power up to 9.0 mW, a threshold determined by the maximum power of the interrogation laser, while consistently maintaining a linear response throughout operation. Due to the low absorption and coefficient of thermal expansion (CTE) of the adhesive (NOA 68, Norland, USA) within the C-Band, minimal opto-thermal effects were observed. Consequently, cavity heating induced by the interrogation laser was minimized, resulting in high stability during operation.”

4. How long did the scanning process for the 2 mm × 2 mm area take?

We have provided the scanning time in the text as follows:

“The total scan time was 28 minutes, primarily necessitated by extensive averaging required to compensate for relative intensity noise (RIN) from the excitation laser. Additional constraints included thermal laser instability and spatial beam cleaning with pinholes, which limited the available laser power to relatively low levels. Utilizing a more stable excitation laser that enables an increase in power while remaining within the ANSI safety limit for single pulses of up to 20 mJ/cm² is expected to reduce the total scan time by a factor of four. Incorporating a galvanometer scanner to guide the excitation beam across the sample could further reduce acquisition times. However, this approach will degrade the SNR at peripheral scan regions due to increased incidence angles. Furthermore, using an excitation laser with a higher pulse repetition rate represents an additional opportunity to further reduce the acquisition time.”

5. The imaging depth seems relatively shallow. Are there any potential methods or modifications you would suggest to increase the imaging depth?

We thank the reviewer for their interest in strategies for improving imaging depth. We have now included this information in the manuscript.

“Imaging can be carried out at greater depths by increasing the excitation power within ANSI safety limits and ensuring homogeneous light delivery through incorporating wide-field illumination that covers the acoustic acceptance field of the detector. Increased sensitivity, reflected by a lower NEPD, is required to detect highly attenuated signals from deep structures. The sensitivity can be further optimized by increasing the optical finesse of the cavity with optimized mirrors and cavity geometries, as well as by improving the pressure phase sensitivity by well-matched acoustic impedances between the water and the polymer resonator and high responsiveness to pressure-induced perturbations of the cavity.”

6. I found the research utilizing multi-core fibers to combine excitation beams and readout particularly interesting. This system is expected to enhance imaging depth—have you measured the extent of this improvement? Additionally, it seems that in vivo measurements with the detector were not conducted. Could you clarify the reason for this?

We thank the reviewer for their interest in the multi-core fiber approach. Due to the close proximity of excitation and ultrasound detector, we anticipate an improvement in SNR and acquisition speed in imaging applications, as suggested by Zhu et al. (2024). The inclusion of the multi-core fiber concept is intended to demonstrate the potential of the TOF-PR design in enabling miniaturization of optoacoustic imaging systems. However, a comprehensive characterization of the probe, including in vivo validation, lies beyond the scope of the current study because the MCF requires a dedicated beam coupling and delivery arrangement, which differs significantly from the current setup. Since the present coupling configuration does not provide high excitation power, additional design and characterization are needed to integrate an efficient fan-out system that increases excitation power. This improvement will make the expected advantages of the MCF more evident and is planned as a continuation of this study. We have made changes to the manuscript as shown below to better reflect these points outlined above.

“As an outlook, we incorporated the TOF-PR design into multi-core fiber arrangements to demonstrate the feasibility of combining ultrasound detection and optoacoustic excitation channels within a single optical fiber (Suppl. Fig.5.). This approach offers a promising and more compact alternative design. We showed that the conical tip of the fiber functioned as an axicon lens, creating a Bessel-like beam. The resulting focal spot has the potential to enable optical-resolution optoacoustic microscopy at sub-millimeter distances, while also performing acoustic-resolution imaging in OptAM at greater depths using diffused light. For a scientifically rigorous in vivo demonstration, a beam coupling arrangement to increase excitation power is necessary. However, the design and characterization of such a system are beyond the scope of this study. We anticipate that this configuration will increase SNR and further drive the miniaturization of optoacoustic imaging systems for space-constrained applications like endoscopy [38]. This expectation is supported by a study by Park et al. (2022), in which a tapered fiber was used to deliver focused light, achieving high-resolution imaging without the need for additional optics [39].”

7. The study by Byullee Park et al. (2022), Shear-Force Photoacoustic Microscopy: Toward Super-Resolution Near-Field Imaging, explores shear-force photoacoustic microscopy using a tapered fiber, which shares similarities with the TOF-PR detector, particularly in leveraging fiber-based architectures for high-resolution optoacoustic imaging. Additionally, the review by Xiaoyi Bao (2023), Prospects on Ultrasound Measurement Techniques with Optical Fibers, provides a broad perspective on fiber-optic ultrasound detection, while Xiaoyi Zhu et al. (2024), High-Speed Innovations in Photoacoustic Microscopy, discusses advancements in high-speed PAM systems, focusing on improvements in temporal resolution and real-time biological imaging. Citing these works could enrich the discussion by providing valuable context on how fiber tapering enhances resolution, minimizes acoustic interference, and optimizes imaging speed, allowing for a more comprehensive evaluation of the TOF-PR detector’s strengths, limitations, and potential improvements.

Thank you for suggesting additional references. We have now incorporated them into our manuscript to emphasize the benefits of fiber tapering.

The study by Park et al. (2022) uses a tapered fiber for light delivery within an optoacoustic setup. This is an interesting and promising approach that could be integrated with a small-footprint detector like the TOF-PR presented in our study to increase imaging resolution and SNR.

“Although in vivo validation was not performed, as it lies beyond the scope of this work, we anticipate that this configuration will increase SNR and further drive the miniaturization of optoacoustic imaging systems for space-constrained applications like endoscopy [38]. This expectation is supported by a study by Park et al. (2022), in which a tapered fiber was used to deliver focused light, achieving high-resolution imaging without the need for additional optics [39].”

We have included the reference by Bao et al. (2023), as shown below, to highlight the importance of high-bandwidth detection. This reference also reinforces the advantages of optical fiber sensors due to their small footprint.

“Systems capturing broadband responses spanning from a few MHz to over 100 MHz have demonstrated superior imaging performance compared to those acquiring bandwidths limited to only a few tens of MHz [2]. This is exemplified by Raster-Scan Optoacoustic Mesoscopy (RSOM), where a high bandwidth enables higher-resolution imaging at shallower depths, facilitating the visualization of fine structures, such as skin vasculature with diameters as small as 7 – 25 μm [3], [4]. Structures extending from the skin surface to the epidermal-dermal junction possess substantial frequency components above 100 MHz, underscoring the necessity for high bandwidth detectors for visualization of fine structures [3], [5].”

In addition, we have cited Zhu et al. (2024) to highlight the advantages of small footprint detectors, which can be better aligned with excitation optics, thereby improving imaging speed and SNR. This reference has been incorporated into our multi-core fiber section to further support the discussion on the alignment of excitation and detection channels.

“This development could lead to a new standard in optoacoustic imaging applications by exploiting small form factor detection, which facilitate alignment with the optical excitation setup. The result is improved SNR and imaging speed in mesoscopic implementations, as well as applications with endoscopic probes and miniaturized biomedical sensors [5], [38].”

“As an outlook, we incorporated the TOF-PR design into multi-core fiber arrangements to demonstrate the feasibility of combining ultrasound detection and optoacoustic excitation channels within a single optical fiber (Suppl. Fig.5.). This approach offers a promising and more compact alternative design. We showed that the conical tip of the fiber functioned as an axicon lens, creating a Bessel-like beam. The resulting focal spot has the potential to enable optical-resolution optoacoustic microscopy at sub-millimeter distances, while also performing acoustic-resolution imaging in OptAM at greater depths using diffused light. Although in vivo validation was not performed, as it lies beyond the scope of this work, we anticipate that this configuration will increase SNR and further drive the miniaturization of optoacoustic imaging systems for space-constrained applications like endoscopy [39].”

Reviewer #3 (Remarks to the Author):

After carefully evaluating the manuscript, I find that while the underlying concept of reducing probe diameter to achieve high bandwidth and mitigate surface acoustic wave (SAW) artifacts is promising, the work presented does not sufficiently demonstrate clear advantages over existing photoacoustic microscopy (PAM) systems. The current findings appear more as

a proof-of-concept rather than a robust, fully developed approach ready for a high-impact journal. Below are my specific concerns:

We appreciate the reviewer's feedback and have carefully addressed all the points raised. We hope that our revised manuscript will now be considered suitable for publication in Nature Communications.

1. Emphasizing an “ultra-small probe” advantage without fully validating its utility

The authors highlight the importance of minimizing the probe diameter and reducing SAW artifacts for high-quality imaging in complex or confined geometries. Yet, the actual experiments—using relatively uncomplicated regions such as mouse ear and dorsal skin—do not fully illustrate how this ultra-small probe would outperform existing PAM systems when targeting truly challenging environments (e.g., curved surfaces, narrow cavities, or hard-to-reach tissue sites).

More representative data should be provided—for instance, imaging the inner ear fold, nail bed, gingiva, or similarly complex geometries. If such testing is not feasible, the Discussion should explicitly clarify how this small probe would outperform established systems in the same application scenarios.

We thank the reviewer for the feedback. As suggested, we have added the following paragraph discussing the advantage of using our detector compared to established detectors. Our current imaging system could be benchmarked against comparable systems. However, direct comparison with PAM systems that operate in challenging environments is not feasible due to our system's operation in trans-illumination mode, which restricts imaging experiments to natural skin folds. This clarification has been incorporated into the text below. In addition, we would like to emphasize that the primary focus of this work is the development of a high-bandwidth ultrasound detector. The PAM system employed was selected to demonstrate the capabilities of this new detector. Further optimization of the PAM system itself lies beyond the scope of the present work.

“By using a polymer external resonator, the resonator compensates for its relatively lower Q-factor compared to SWEDs or π -phase shifted Bragg gratings by using materials with superior acoustic properties, resulting in better acoustic impedance matching and higher acoustic phase sensitivity. In contrast to established polymer-detectors, this developed detector achieved high sensitivity and bandwidth comparable to silicon detectors. Moreover, the adverse effects of SAWs are mitigated through the miniaturization of the detector's effective size and the use of polymeric materials. This advancement enables performance similar to silicon detectors with a substantially smaller footprint, which facilitates high-resolution imaging in space-constrained applications like endoscopy. However, imaging demonstrations in more complex regions is currently not possible due to the current imaging system's operation in trans-illumination mode, which limited experiments to natural skin folds.”

2. Lack of direct quantitative comparisons with existing PAM systems

Despite repeatedly citing the advantages of a miniaturized footprint, high bandwidth, and reduced SAW artifacts, the manuscript does not present a side-by-side comparison with other high-frequency PAM systems or fiber-optic transducers under matching conditions. Many reported techniques exist with bandwidths at or above 100 MHz, making direct comparisons (in terms of resolution, SNR, and acquisition speed) essential to justify claims of superiority.

Conduct or include head-to-head imaging comparisons between the proposed probe and at least one established high-frequency PAM (or fiber-based Fabry-Pérot) system. Measures

such as SNR, resolution, field of view, and data acquisition time under identical conditions would clarify the true performance gains.

We thank the reviewer for their suggestion. In response, we have included a table comparing our detector with a high-frequency PAM system, that offers comparable resolution and bandwidth. To the best of our knowledge, no fiber-optic Fabry-Pérot has achieved comparably high resolution and bandwidth. Therefore, we have included a silicon waveguide etalon detector (SWED), which would be a more suitable reference detector for comparison.

The table presents the SNR and resolution of the respective detector. To allow for an SNR comparison that is independent of the acoustic source intensity, we report the noise equivalent pressure density (NEPD), which quantifies the minimum detectable pressure over a specified measurement bandwidth. The NEPD corresponds to an SNR of 1 across the bandwidth used for the characterization. Metrics such as field of view and data acquisition time are not included, as they are heavily dependent on the specific imaging system and do not solely represent the properties of the ultrasound detector.

	Dimensions		Bandwidth		Sensitivity		Resolution	
	Footprint	Thickness [μm]	-3 dB [MHz]	-6 dB [MHz]	NEPD [$\text{mPa}\cdot\text{Hz}^{-1/2}$]	NEPD x Area [$\text{mPa}\cdot\text{mm}^2\cdot\text{Hz}^{-1/2}$]	Lateral [μm]	Axial [μm]
TOF-PR	\emptyset 24 μm	6	110	149	1.5 (25 MHz)	9.0×10^{-4}	17	7
[23]	\emptyset 125 μm	21	25	30	11 (25 MHz)	1.3×10^{-1}	> 38*	> 35*
[24]	\emptyset 125 μm	16	20	25	2.1 (20 MHz)	2.5×10^{-2}	> 47*	> 45*
[25]	\emptyset 125 μm	20	-	30	40 (30 MHz)	4.9×10^{-1}	84	231
[22]	\emptyset 125 μm	30	25	-	70 (25 MHz)	4.3×10^2	> 37*	>36*
[16]	3000 μm x 800 μm	9	-	230	9 (25 MHz)	9.9×10^{-7}	16	5

“The superior sensitivity and resolution of the TOF-PR are evident through direct comparison with state-of-the-art ultrasensitive extrinsic polymer resonators [25],[26],[27], as well as to an intrinsic SWED [16], as summarized in Table 1. Although extrinsic polymer detectors have comparable footprints to the TOF-PR, their substantially lower resolution and sensitivity limit their utility for high-resolution imaging in challenging environments, such as endoscopic applications. Conversely, SWED offers a considerably higher bandwidth, but TOF-PR can achieve comparable axial and lateral resolutions by reducing SAW interference. Moreover, TOF-PR fabrication only requires a wet lab environment, which reduces manufacturing time and cost by approximately 100-fold.”

3. Point-scanning time constraints remain unaddressed

Like most high-resolution acoustic or photoacoustic methods, the proposed system uses raster scanning. The manuscript does not detail frame rates, sampling speeds, or strategies to handle motion artifacts during in vivo imaging. This limits the scope of practical applications, especially for larger fields of view or scenarios requiring real-time insights.

Provide explicit information on scanning speed and potential methods to accelerate data

acquisition or mitigate motion artifacts. For instance, an array-based or combinatorial scanning method might alleviate the inherent speed limitations of raster scanning.

We thank the reviewer for their suggestion. In response, we have provided the requested information on scan time and have also discussed potential strategies for accelerating data acquisition. These strategies for accelerating scan speed are also expected to reduce the risk of motion artefacts.

“The total scan time was 28 minutes, primarily necessitated by extensive averaging required to compensate for relative intensity noise (RIN) from the excitation laser. Additional constraints included thermal laser instability and spatial beam cleaning with pinholes, which limited the available laser power to relatively low levels. Utilizing a more stable excitation laser that enables an increase in power while remaining within the ANSI safety limit for single pulses of up to 20 mJ/cm² is expected to reduce the total scan time by a factor of four. Incorporating a galvanometer scanner to guide the excitation beam across the sample could further reduce acquisition times. However, this approach will degrade the SNR at peripheral scan regions due to increased incidence angles. Furthermore, using an excitation laser with a higher pulse repetition rate represents an additional opportunity to further reduce the acquisition time.”

The question regarding the use of an array-based approach is similar Reviewer 1’s fourth question. For ease of reference, we have pasted our response below.

“An array-based approach was not selected for this study due to the requirement for individual tuning of each detector to its quadrature point. Nevertheless, the detector design and functionality would allow such an array configuration. However, implementing this approach would require multiple interrogation sources to tune each element individually, thereby increasing the complexity of the read-out circuit. Another method is the use of almost identical low-finesse elements that can be operated at the same wavelength [22], [37].”

4. “Multicore fiber” work restricted to proof-of-concept

While the integrative approach for excitation and detection in a single multicore fiber is intriguing for minimally invasive or endoscopic applications, it is only demonstrated in a rudimentary manner. Critical performance indicators—such as achieving uniform illumination, effective coupling efficiency, and imaging depth in biologically relevant conditions—are either not reported or only superficially addressed.

If the authors consider endoscopy a key selling point, they should substantiate it with more concrete data using realistic phantoms or biological tissues. A deeper discussion of coupling efficiencies and beam uniformity is also necessary to solidify claims of practical endoscopic potential.

We thank the reviewer for their comment regarding the multi-core fiber approach. Our objective in this manuscript is to introduce the integration of the TOF-PR into a multi-core fiber configuration and to highlight the design’s great potential for miniaturizing optoacoustic imaging systems. This potential arises from the close spatial arrangement of excitation and ultrasound detection channels, which is expected to improve SNR and acquisition speed in imaging applications, as supported by Zhu et al. (2024). Additionally, we demonstrated that the conical tip of the fiber acts as an axicon lens and creates a Bessel-like beam. The resulting focal spot may enable optical-resolution optoacoustic microscopy at sub-millimeter distances, while also performing acoustic-resolution imaging in OptAM using diffused light at greater depths. However, these advantages are currently limited by the lack of fully optimized beam coupling and delivery arrangement for high optical power. The development

and characterization of such a system extends beyond the scope of this work and will be addressed in a future study.

We anticipate that the multi-core fiber configuration will increase SNR and further promote the miniaturization of optoacoustic imaging systems. This miniaturization, enabled by the TOF-PR design, will without a doubt make systems highly suited for future use in space-constrained applications, such as endoscopy. While additional experiments on phantoms and tissues will be essential to facilitate translation of this technology to endoscopic applications, such investigations are beyond the scope of the current manuscript. The focus of this work is to present a significant advancement in the development of a high-bandwidth ultrasound detector for optoacoustic imaging.

5. Gap between theoretical advantages and practical feasibility

Although the polymer resonator design and considerations for reducing SAW artifacts are theoretically commendable, the discussion around reproducibility, mechanical stability, and durability of the device (particularly in vivo or clinical environments) is insufficient. The minimal demonstrations provided make it difficult to gauge readiness for clinical or commercial usage.

For publication in a high-impact journal, more rigorous demonstrations of packaging, mechanical reliability, and performance in complex real-tissue scenarios are strongly recommended. At a minimum, the authors should address how they plan to overcome obstacles in manufacturing, device longevity, and quality assurance.

We thank the reviewer for their comment and acknowledge that Nature Communications is a high-impact journal. However, the journal's stated aim is to "represent important advances of significance to specialists within each field", such as engineering sciences. In this context, we believe our manuscript is suitable for consideration, as it presents a significant advance in high bandwidth ultrasound detector development for optoacoustic imaging that will be of interest to specialists in the field. While we agree that more rigorous demonstrations of packaging and manufacturing, mechanical reliability, performance in complex real-tissue scenarios, and quality assurance are important for eventual clinical or commercial translation, we believe that these aspects are beyond the scope of this study. At this stage, our focus is on demonstrating a major technological advance, which aligns with journal's objectives. Nevertheless, we have added a section that discusses the prospects of the TOF-PR detector in the context of manufacturing and durability, and quality assessment during production.

"The established fabrication process for the TOF-PR detector features strong potential for automation using CNC processes and robots. In particular, automation will ensure the precise and consistent formation of the epoxy cavity, which is highly dependent on tight control of production parameters to achieve optimal detector performance. During all fabrication steps, the reflection spectrum of the detector can be analyzed to identify faulty detectors. For future in vivo applications, a well-established method that can be applied involves protecting the detector with a thin polymer layer on the outer mirror to guard against mechanical influences and corrosion [29], [30]."

Overall Summary and Recommendation

In its current form, the manuscript does not convincingly demonstrate a transformative step beyond existing high-frequency PAM approaches. The core claims—namely, that an ultra-small, high-bandwidth probe offers significant benefits for complicated or endoscopic imaging—remain mostly speculative, as the data presented focus on relatively simple, flat tissue sites and limited proof-of-concept fiber integration.

Therefore, based on the present scope and evidence, I believe the manuscript's novelty and completeness are not sufficient for publication in Nature Communications. The authors are encouraged to perform additional, more rigorous experiments—especially in complex geometries or genuine endoscopic scenarios—and to provide robust comparisons with state-of-the-art systems. Addressing the practical engineering, reproducibility, and scanning-speed limitations would also greatly enhance the impact and applicability of the work.

We have carefully addressed the reviewer's concerns regarding the readiness of the detector for clinical application and demonstration of its performance in challenging environments. Additionally, we have clarified details regarding the scan durations and included a comparison of the TOF-PR to a high-bandwidth ultrasound detector. Furthermore, we expanded the discussion on the multi-core fiber approach, noting that extensive experimental validation is beyond the scope of this work.

We would like to reiterate that our primary aim, which aligns with the aim of Nature Communications, is to present a significant advance in detector development that will be of interest to specialists in the field. This work closes the gap between high-bandwidth optical ultrasound detectors with large footprints and the well-established extrinsic Fabry-Pérot designs on the tips of optical fibers, which have until now encountered limitations in imaging resolution and sensitivity. The TOF-PR ultrasound detector advances the miniaturization of high-bandwidth ultrasound detectors to a new level, making it well-suited for space-constrained applications, such as endoscopy, while delivering superior resolution and sensitivity. At this stage, presenting a product ready for commercial implementation, including comprehensive studies on packaging, manufacturing, mechanical reliability, and quality assurance, is beyond the scope of this manuscript. While the ultimate goal is translation into clinical applications, such as endoscopy, this process will require additional time and development. Our manuscript represents a crucial step toward that goal, as the combination of high bandwidth and resolution while maintaining a small form factor, makes the detector highly promising. Regarding the other comments, we have ensured that we have expanded the discussion to include comparisons to PAM approaches and provided projections on the detector's performance in challenging scenarios. We hope these major revisions adequately address the reviewer's concerns and allow our manuscript to be considered for publication in Nature Communications.

Reviewer #4 (Remarks to the Author):

The manuscript titled "150 MHz Polymer Resonator for Optoacoustic Mesoscopy based on a Tapered Optical Fiber" presents a tapered optical fiber polymer resonator (TOF-PR) with a 6 μm thick polymer resonator, achieving a 150 MHz detection bandwidth and a noise-equivalent pressure density (NEPD) of $1.5 \text{ mPa}\cdot\text{Hz}^{-1/2}$, which shows a broader bandwidth and lower NEPD than previous works [1-5]. The authors claim that the miniaturization of the fiber tip and the subsequent fabrication of a thinner Fabry-Perot (FP) cavity represents a significant advancement in optoacoustic detection. However, upon thorough comparison with existing literature, this work lacks the novelty and scientific depth required for publication in Nature Communications.

We appreciate the reviewer's feedback and have carefully addressed all the points raised below. We hope that our revised manuscript will now be considered suitable for publication in Nature Communications.

Major Concerns and Recommendation for Rejection

Limited Novelty and Incremental Contribution: The concept of enhancing bandwidth by reducing the FP cavity thickness was already disclosed in the 2017 Nature Photonics paper [1]. The current manuscript applies this idea by tapering the fibre tip to fabricate a thinner cavity, claiming this as a novel, more accessible fabrication method. However, tapering the fiber tip is not a new strategy—this approach has been thoroughly explored in Zhang et al. (2015) [6] and subsequent works. Thus, this manuscript essentially replicates previously established strategies, making the improvements in acoustic characteristics and bandwidth expected outcomes rather than novel contributions.

We thank the reviewer for pointing out that the novelty and contribution of our work were not stated clearly enough. We acknowledge that previous studies have attempted to reduce FP cavity thicknesses and modify the tip. However, these efforts had limitations that prevented them from fully leveraging the potential benefits of optimizing the detector's cavity geometry for superior acoustic and optical properties.

For example, Guggenheim et al. (2017) demonstrated that reducing cavity size could increase bandwidth. However, the radius of the curvature of their detector's tip could not be adjusted independently of the cavity length, preventing the selection of the ideal combination for high bandwidth and high finesse, thereby limiting overall performance. Similarly, Zhang et al. (2015) flattened the frequency response by minimizing acoustic diffraction. However, despite rounding the tip, the detector's aperture was identical to the fiber diameter, which compromised the optical confinement, resulting in reduced optical phase sensitivity as the cavity length decreased. In addition, due to the large aperture of the detector, the detector is expected to be prone to SAW interference. To clarify the distinction between our detector design and those in the literature, as well as to emphasize the transformational advance achieved, we have updated our text accordingly:

"A reduction in cavity size, and consequently an increase in bandwidth, has been demonstrated on glass substrates for sensors with apertures as small as $390 \mu\text{m}$ in diameter [26]. However, the radius of curvature could not be adjusted independently from cavity length due to the surface energy interactions between the substrate and the epoxy. Thus, reducing the cavity length to increase bandwidth compromises optical confinement within the cavity, leading to decreased sensitivity. In addition, this method is unsuitable for cavity formation on fiber tips, since the large apertures would exceed the $125 \mu\text{m}$ diameter of a standard single-mode fiber."

“Attempts have been made to flatten the frequency response by minimizing acoustic diffraction by rounding the fiber tip [28]. However, optical confinement was not optimized, and the detector’s aperture was identical to the fiber diameter, making the detector less sensitive and more susceptible to SAW interference.”

Redundancy with Prior Work: The methods used in this study are redundant with prior works. Zhang et al. (2015) [6] demonstrated the optimization of fiber-optic ultrasound sensors through fiber tip modifications, including tapering and cavity design. The Optics Express publications by Marques et al. (2020) [7], Martin-Sanchez et al. (2022) [8], and Martin-Sanchez et al. (2023) [9] provide comprehensive models and experimental validations of Fabry-Perot etalons illuminated by focused beams, Gaussian beam propagation in FP etalons, and plano-concave microresonator behaviour. These studies cover the theoretical foundations and experimental optimizations that this manuscript overlooks, further highlighting its lack of originality.

We thank the reviewer for their comment regarding the need to more clearly highlight the novelty and originality of this work.

As mentioned in our response to comment 1 from reviewer 4, Zhang et al. (2015) achieved a flattened frequency response by minimizing acoustic diffraction using a rounded fiber tip. However, their cavity was not optimized in terms of shape, which negatively affected the optical back-coupling efficiency. In contrast, the TOF-PR design presented in this work overcomes the trade-off between acoustic properties and optical back-coupling efficiency by enabling independent adjustment of cavity curvature and size. The tapered shape effectively eliminates diffraction at the tip, which resulted in a flat frequency response and reduced SAW interference.

Marques et al. (2017) and Martin-Sanchez et al. (2022) investigated the optical properties of Fabry-Pérot etalons and simulated them. However, these studies focused on flat Fabry-Pérot resonators that do not have a curved cavity design. Hence, these models account for beam walk-off, which reduces optical phase sensitivity. However, this limitation is not present in the TOF-PR due to its precisely adjusted radius of curvature of the cavity. While Martin-Sanchez et al. (2023) included the curvature of the cavity in the ABCD-Matrix model and investigated different Gaussian beam geometries, and their influence on the interferometer transfer function, their model validation relied on a setup using free beam optics to change the beam characteristics. This approach does not translate directly to polymer resonator detectors built on fiber tips like the TOF-PR, where the beam shape is defined by the employed optical fiber and the optical properties of the polymer.

In this manuscript, the shape of the cavity and mirror reflectivities are optimized to maximize optical phase sensitivity. Furthermore, the TOF-PR design is optimized for acoustic detection and bandwidth by comparing different cavity sizes in terms of their bandwidth and achievable resolution. Importantly, the acoustic properties, which are critical for an effective optical ultrasound detector, are not addressed in Martin-Sanchez et al. (2023), making our study distinct in its approach.

Neglect of Optical and Acoustic Mechanisms: The manuscript reports a reduction in surface acoustic wave (SAW) artifacts but provides no rigorous physical analysis or modeling to explain this phenomenon. The authors simply observe SAW suppression without exploring the underlying acoustic mechanisms or optimizing the design parameters. Similarly, the optical interrogation of the tapered fiber significantly affects the optical field distribution in FP Cavity, yet the manuscript lacks an analysis of how tapering impacts wavefront propagation, cavity finesse, or sensitivity.

Absence of Design Optimization: Unlike the referenced works, particularly Zhang et al.

(2015) [6], which conducted comprehensive design optimizations, this manuscript does not explore the influence of taper angles, polymer thicknesses, or cavity curvatures on device performance. The lack of parametric studies indicates a superficial approach to the research, falling short of the scientific rigor expected for publication in Nature Communications.

We thank the reviewer for their comment. In response, we have added a detailed explanation of how reducing the detector size contributes to the reduction of SAWs. Furthermore, we added a statement to clarify that the base diameter of the taper (PR-III: 24 μm) is much larger than the mode field diameter (MFD) of the fiber used (MFD of PM1550-XP: 10.1 μm). Hence, no light propagates within the tapered area, resulting in no impact on wavefront propagation and, therefore, no loss in finesse and sensitivity.

“SAWs are excited at the solid-liquid interface when longitudinal waves (L-waves) are incident at angles near the Rayleigh angle [31]. In sensors with a larger base diameter, the total sensor area exceeds the ultrasound-sensitive area, increasing susceptibility to SAW excitation. By reducing the size of the polymer resonator to closely match the effective sensitive area, the impact of SAWs is minimized. The taper is therefore essential to eliminate the unused part of the optical fiber, which would otherwise act as a source of SAWs [16]. Due to the orientation of the taper, the Rayleigh criterion is not met at the silica-water interface in this area, effectively suppressing SAW generation. Furthermore, the taper does not interfere with beam propagation, as the minimum taper diameter is more than twice the mode field diameter (MFD) of the fiber. Consequently, no relevant amount of light is carried within the tapered area of the fiber.”

Regarding design optimization, the radius of curvature was adjusted to achieve maximum finesse for all three polymer resonators. We investigated the effects of reducing the base diameter and cavity length, while maintaining optical confinement, on the frequency response and resolution. The results for PR-I, PR-II and PR-III with cavity thicknesses of 35 μm , 10 μm and 6 μm , respectively, are shown in Figures 2 a-d. The taper effectively suppresses SAWs across a broad range of taper angles due to their orientation relative to the Rayleigh angle of incoming L-waves. The specific taper angle is primarily determined by the mechanical stability of the tip during operation and the constraints of the fabrication process. The comprehensive consideration of all relevant parameters in the TOF-PR design, along with the successful optimization of PR-III through adjustment of the base diameter and the cavity thickness, demonstrates a thorough scientific analysis aimed at meeting the standards for publication in Nature Communications.

Overall Poor Completion of Study: The study appears to be an incomplete extension of the ideas presented in 2017 [1], with minimal modifications to the fiber tip geometry and no additional theoretical or experimental advancements. The lack of depth in both optical and acoustic analyses reflects an underdeveloped study that does not meet the standards of a high-impact journal.

Conclusion and Recommendation: In conclusion, this manuscript lacks the originality, depth, and scientific rigor required for publication in Nature Communications. The primary contribution—tapering the fiber tip to fabricate a thinner FP cavity—is an incremental improvement that does not justify publication in a high-impact multidisciplinary journal. The absence of theoretical analysis, optimization, and comprehensive comparison with existing literature further weakens the manuscript.

Given these limitations, I strongly recommend rejection and suggest submission to a

specialized optics journal where this work may be more appropriately categorized as a Technical Note.

We thank the reviewer for their valuable feedback and have carefully considered the concerns raised in our revision of the manuscript. While we understand the concern regarding the perceived limited novelty and contribution, we would like to emphasize that the central aim of our study is to present a significant advancement in the development of miniaturized, high-performance optical ultrasound detector development. This new detector enables high-resolution imaging in space-constrained applications, such as endoscopy, representing a major step forward in the overall miniaturization of optoacoustic imaging and sensing technologies.

Our work addresses a critical gap between large-footprint, high-bandwidth optical detectors and fiber-tip Fabry–Pérot designs, which have so far been constrained by limited sensitivity and spatial resolution. To the best of our knowledge, no previously reported device achieves the same combination of axial and lateral resolution, bandwidth, and sensitivity within such a compact footprint.

We also acknowledge the reviewer's point regarding the insufficient discussion on the underlying principles of SAW generation and its mitigation. In response, we have substantially revised the manuscript to provide a more in-depth explanation of the physical origins of SAW excitation at solid–liquid interfaces and we now detail how the taper geometry contributes to effective SAW suppression.

Regarding the concern that the study appears incomplete, we would respectfully like to clarify that this work presents a comprehensive investigation into the design and optimization of an optical ultrasound detector, grounded in the fundamental principles of extrinsic Fabry–Pérot interferometry. It demonstrates significant improvements in both optical and acoustic performance and includes in vivo imaging experiments that highlight the detector's practical capabilities and suitability for optoacoustic mesoscopy applications.

We hope these substantial revisions and clarifications sufficiently address the reviewer's concerns and underscore the importance of this work, supporting its suitability for consideration for publication in Nature Communications.

References

1. Guggenheim, James A., Jing Li, Thomas J. Allen, Richard J. Colchester, Sacha Noimark, Olumide Ogunlade, Ivan P. Parkin et al. "Ultrasensitive plano-concave optical microresonators for ultrasound sensing." *Nature Photonics* 11, no. 11 (2017): 714-719.
2. Li, Guangyao, Zhendong Guo, and Sung-Liang Chen. "Miniature all-optical probe for large synthetic aperture photoacoustic-ultrasound imaging." *Optics express* 25, no. 21 (2017): 25023-25035.
3. Chen, Bohua, Yuwen Chen, and Cheng Ma. "Photothermally tunable Fabry–Pérot fiber interferometer for photoacoustic mesoscopy." *Biomedical Optics Express* 11, no. 5 (2020): 2607-2618.
4. Yang, Liuyang, Dongchen Xu, Geng Chen, Anqi Wang, Liangye Li, and Qizhen Sun. "Miniaturized fiber optic ultrasound sensor with multiplexing for photoacoustic imaging." *Photoacoustics* 28 (2022): 100421.
5. Lin, Wei-Kuan, Linyu Ni, Xueding Wang, Jay L. Guo, and Guan Xu. "Fabrication of a translational photoacoustic needle sensing probe for interstitial photoacoustic spectral analysis." *Photoacoustics* 31 (2023): 100519.
6. Zhang, Edward Z., and Paul C. Beard. "Characteristics of optimized fibre-optic ultrasound receivers for minimally invasive photoacoustic detection." In *Photons Plus Ultrasound: Imaging and Sensing 2015*, vol. 9323, pp. 151-159. SPIE, 2015.

7. Marques, Dylan M., James A. Guggenheim, Rehman Ansari, Edward Z. Zhang, Paul C. Beard, and Peter RT Munro. "Modelling Fabry-Pérot etalons illuminated by focussed beams." *Optics Express* 28, no. 5 (2020): 7691-7706.
8. Martin-Sanchez, David, Jing Li, Dylan M. Marques, Edward Z. Zhang, Peter RT Munro, Paul C. Beard, and James A. Guggenheim. "ABCD transfer matrix model of Gaussian beam propagation in Fabry-Perot etalons." *Optics Express* 30, no. 26 (2022): 46404-46417.
9. Martin-Sanchez, David, Jing Li, Edward Z. Zhang, Paul C. Beard, and James A. Guggenheim. "ABCD transfer matrix model of Gaussian beam propagation in plano-concave optical microresonators." *Optics express* 31, no. 10 (2023): 16523-16534.

Response to reviewers for the manuscript "150 MHz Polymer Resonator for Optoacoustic Mesoscopy based on a Tapered Optical Fiber"

Reviewer 3

I have reviewed the revised manuscript and the authors' response to my previous critiques. While I acknowledge the effort taken to address some of the points, my fundamental concerns regarding the novelty, transformative potential, and practical validation of this work remain largely unresolved. The core issue is that the manuscript presents a solid engineering optimization within a known design space, but it fails to demonstrate a conceptual leap or a clear, validated advantage that would constitute a significant advance for the broad readership of Nature Communications.

The authors' responses often justify the absence of key experiments by stating they are "beyond the scope of this study" (Line 410). However, for a journal of this stature, precisely those experiments, such as validation in challenging anatomical environments or a direct, system-level performance comparison, define the scope required to substantiate the high-impact claims being made. Below are a few of my major persistent concerns.

First, the "Ultra-Small Probe" advantage remains largely speculative and untested. The authors argue that the miniaturized footprint is key for space-constrained applications like endoscopy (e.g., Lines 51, 412). However, the imaging demonstrations remain confined to simple, flat, and accessible surfaces (mouse ear, dorsal skin). The rebuttal concedes that imaging in complex regions is "not possible" with their current trans-illumination system. This is a critical omission. A claim of endoscopic utility is severely weakened without data from a curved, confined, or otherwise challenging geometry that would actually benefit from the small size. The provided data does not prove that this probe enables something that existing, larger probes cannot achieve in these same, simple scenarios.

We would like to thank the reviewer for raising this concern, as it gives us the opportunity to further elaborate on the status of the field and the importance of the development in the current paper reviewed. We listed endoscopy as one of future directions, since this has been a key motivator in major prior works; therefore, omitting mention of it would probably raise questions from other reviewers. In fact, our paper follows the rationale of other major publications, i.e:

- *Guggenheim et al. Nature Photonics 11, 714-719 (2017)*
- *Shnaiderman et al. Nature 585, 372-378 (2020)*
- *La et al. Nature Communications 15, 7521 (2024)*
- *Westerveld et al. Nature Photonics 15, 341-345 (2021)*
- *Hazan et al. Nature Communications 13, 1488 (2022)*

These studies claimed endoscopy and miniaturized applications as future possibilities but performed validation in NON-constrained environments using freely accessible geometries such as the mouse ear and artificial phantoms!

Future directions do not necessarily need to be demonstrated within a single paper; otherwise, none of the aforementioned papers would have been published by Nature journals. We are certain that the reviewer fully appreciates that science does not work by demonstrating technology from technology readiness level (TRL) 1 to TRL 6 in a single publication. For example, it took at least a decade before the first optical coherence tomography (OCT) paper led to meaningful applications. Similarly, despite its might and top impact publications, Raman spectroscopy has yet to find routine biomedical applications. Achieving high bandwidth in ultrasound detection, using a fiber sensor, as demonstrated herein for the first time from TRL 1 to TRL 3-4, is a notable feat in itself.

It is widely recognized that progressing through TRLs can take more than 1-2 years and may involve expenditures of €1-2 million per TRL. Therefore, the development timelines and costs for a detector and an endoscope may differ by an additional 3-4 years and €5-6 million. Likewise, scientific publications, such as the ones listed above, can communicate an idea or a new design with adequate experiments that support the new development, not full demonstration in future operational environments, i.e. at a high TRL.

*Nevertheless, it is not endoscopy but **the performance of the detector presented herein that cannot be matched by any other detector currently available**, which forms the basis of this publication. This performance leads to additional pragmatic advantages that have nothing to do with endoscopy:*

In relation to other fiber sensors:

- *No other fiber detector to date has demonstrated such **300% bandwidth improvement**, reaching the 150 MHz range, and minimization of SAW. Consequently, the optoacoustic performance demonstrated herein has never been achieved before.*
- *No other fiber detector has achieved such bandwidth improvement **without loss of sensitivity**, i.e. without a trade-off, which is not an anticipated finding.*
- *The paper further demonstrates a **uniform bandwidth response**, despite the 300% bandwidth improvement over other fiber sensors, also not an anticipated finding and the result of the new design introduced.*

In relation to piezo-electric (PZT) detectors:

- *No PZT of such detector area size can achieve such bandwidth and sensitivity.*
- *Large PZT's block the optoacoustic illumination, thus hindering effective implementations. Therefore, this is the first detector that can effectively deliver*

150MHz bandwidth without blocking the illumination, which can lead to new handheld sensor designs, not endoscopes.

*The effective gain in size of the fiber detector presented in our paper is at least 30-times over PZT detectors of same bandwidth and sensitivity, **300% better in bandwidth than any previously presented fiber detector**. These are critical and undeniable advantages of the fiber detector we present, for the entire field of ultrasound detectors. **This advance is independent of whether this fiber detector will be used in optoacoustic sensing of the skin and other non-invasive applications or as an endoscope.***

For disambiguation, we now discuss in the paper the application of this fiber detector as an advantage in optoacoustic sensors (not only endoscopes), where the small size and high bandwidth can allow new sensing capabilities and removed the statement on imaging in complex geometries. These new capabilities are due to the small size, which will allow an illuminator to be very close or concentric to the detector and the high bandwidth, which leads to 7 micrometer resolutions as a function of depth never before possible by other fiber detectors demonstrates so far.

“The design presented in this study successfully combines high sensitivity with uniform 150 MHz broadband ultrasound detection. Until now, this has typically required large PZT transducers, which limit probe miniaturization by blocking optoacoustic illumination. Consequently, our detector enables new space-constrained optoacoustic applications, such as miniaturized sensors or endoscopy.”

Second, in my personal opinion the work represents incremental advance, but not a paradigm shift. The authors have more clearly articulated their position that the TOF-PR combines the benefits of silicon detectors (bandwidth) and polymer detectors (acoustic coupling). However, this is presented as an engineering trade-off and optimization, not a fundamental breakthrough. The core concepts using a polymer for better acoustic impedance, reducing cavity size for bandwidth, and tapering a fiber tip are previously established, as rightly noted by other reviewers. The authors' work refines these concepts but does not introduce a new principle or mechanism. The performance improvements, while commendable, appear to be an expected outcome of this refinement process rather than a surprising or disruptive discovery. The manuscript reads as a good, specialist engineering paper, but not one that redefines the field's boundaries.

The core of this comment relates to what is an “engineering optimization” and an “incremental advance” vs. a foundational advance.

*Our detector offers, a **300% improvement in acoustic bandwidth** over any prior fiber optic polymer resonator published, reaching an unprecedented >150 MHz bandwidth. This advance corresponds to a three-fold improvement in resolution. In addition, SAW is virtually eliminated and the uniformity of the frequency response presented is exemplary and superior to previous demonstrations (see Fig. A1 comparing to the Guggenheim paper).*

The inventive detector herein achieves the **300% bandwidth improvement at no loss of sensitivity, i.e. not a trade-off**, over previously published detectors. Therefore, this is not a trade-off but a real foundational advance in the field.

Advances of this magnitude have historically been recognized as significant in biomedical imaging. **Optimization** typically means a process that better aligns, calibrates, tunes etc of a system, to **improve performance by 10-20%**. A new design leading to 300% improvement is by all definitions a fundamental advance, not an optimization.

Critically, the marked performance advance of our detector is based on a **new, non-obvious design**, since it has never been disclosed or characterized before. The key advance for achieving such performance is the ability to miniaturize the cavity to only 6 micrometres dimension on top of a fiber. The geometry of the cavity is controlled with precision by a taper to achieve **high-quality resonance and frequency response uniformity**. There is no publication or prior art that indicates that this is possible or that there will be frequency uniformity or no-loss of sensitivity.

In fact, one can argue the contrary: **established fiber optic Fabry-Pérot ultrasound detectors have not demonstrated the ability to increase bandwidth without compromising frequency response uniformity.**

We acknowledge that high bandwidth and uniform frequency response have been demonstrated previously, from a different design, i.e. by silicon waveguide etalon detectors (SWED) (see our Shnaiderman et al. Nature 2020). However, the SWED results clearly show that high bandwidth and spectral uniformity alone do not necessarily translate into high imaging resolution. The achievable resolution was reduced by the coupling of surface acoustic waves, which introduced imaging artefacts and broadened the point spread function (PSF). These observations highlight that suppressing SAWs is just as important for achieving optimal imaging quality.

The TOF-PR takes SAW suppression into consideration in its fiber optic detector design. By accounting for the Rayleigh excitation criterion that governs SAW generation and reducing the detector aperture to match the sensitive detection area, the coupling of SAWs into the detection region is markedly minimized. As a result, this approach leads to substantial improvement in imaging resolution, almost comparable to that achieved by SWEDs, without the need for such high bandwidths and complex manufacturing methods. Thus, the advancements yielded by the TOF-PR design in optoacoustic imaging are not solely reflected by metrics, such as bandwidth and frequency response uniformity.

Improvements of this scale (**300% on bandwidth**) are not called incremental or obvious. For instance, anecdotally, if such advance was incremental, the 2014 Nobel Prize awarded for super-resolution photo-activated localization microscopy (PALM) could be viewed, in retrospect, as “just” an “obvious” fluorescence source localization using the photo-activation principle known since the 1990’s. PALM for example

improved the combined resolution (axial and lateral) by a similar 300% combined (Axial and Lateral resolution) over 4Pi microscopy – available since the 1990's.

Third, confusion of in vivo and ex vivo. On Lines 362-363, it says “To demonstrate the performance and resolution achieved in-vivo (Suppl. Fig.4), we scanned a 2 mm x 2 mm scan area at the mouse ear, with a scan step of 5 μ m (Fig.4b).” But according to the figure caption of Figure 4, the results are from a mouse ear ex vivo.

We thank the reviewer for the attentive reading. The in-vivo experiment relates to Supplementary Figure 4, which demonstrates in-vivo imaging on mouse dorsal skin, while Figure 4 demonstrates the ex-vivo mouse ear imaging experiment. We have restructured the sentence to avoid confusion.

“To demonstrate the performance and achievable resolution, we scanned a 2 mm x 2 mm scan area at the mouse ear, with a scan step of 5 μ m (Fig.4b).”

In summary, the revisions have improved the manuscript's clarity, but the work is an incremental development; the performance demonstrated is strong, but the claim of transformative potential for endoscopy is still not supported by the necessary experimental validation in relevant, challenging environments. For these reasons, I think the work would be more appropriately suited for a leading, but more specialized, journal in the field of optical or ultrasound sensing.

We thank the reviewer for their assessment, which we have addressed comprehensively above. We present a new design which leads to a major improvement of performance. A

- **300% improvement in bandwidth**
- **a uniform bandwidth distribution,**
- **a minimization of SAW,**
- **at no loss of sensitivity**

are not incremental or obvious performance and have never been presented before. Likewise, no design has previously shown capable of achieving such performance.

On the premise of personal opinions on what is obvious or incremental, in retrospect, everything is obvious. We mentioned the super-resolution microscopy anecdote above. Similar obvious in retrospect are the fields of micro-fluidics, nanotechnology etc, in the sense that miniaturization is always predictable. But it is the particular technology that allows the miniaturization that bears the weight of the invention. We also note that in science, prediction and experimental demonstration are not equal: Einstein predicted gravitational waves, but it took 60 years and another Nobel prize to experimentally demonstrate them by Hulse and Taylor.

Of course, we do not argue that our work is anywhere close to a level that can be compared with these colossal scientific developments. What we argue is that “obvious” and “incremental” does not match the unprecedented performance advances offered

by our new miniaturized cavity design based on a tapered optical fiber. Likewise, we argue that a “prediction” is very far from an “experimental demonstration”, because it requires a real design that actually works experimentally, which is non-obvious.

#Reviewer 5

After carefully checking the revision, rebuttal, and the cited prior art, I believe the manuscript shows substantial conceptual and implementation overlap with Guggenheim et al., Nat Photonics (2017), while the central claimed advance— independent control of the tip radius of curvature (ROC) and cavity length (L)—is not substantiated by data. Below are the key points.

1) Misrepresentation of prior art (fibre-tip feasibility)

The manuscript states that the prior method is unsuitable for fibre tips because the required aperture would exceed the 125 μm diameter of a standard single-mode fibre. This is factually incorrect: Guggenheim (2017) explicitly demonstrated plano-concave microresonators on single-mode fibre tips (125 μm cladding), with $L = 16 \mu\text{m}$ (~40 MHz) and $L = 12 \mu\text{m}$ (~55 MHz) devices and near-omnidirectional acceptance. The current text should be corrected, and the comparison table should be updated accordingly. As written, the manuscript risks selectively understating prior fibre-tip results.

We thank the reviewer for bringing this concern to our attention. We have updated the introduction to clarify this issue, along the points that follow.

Our detector achieves

- **150 MHz in bandwidth,**
- **a uniform bandwidth distribution,**
- **a minimization of SAW,**
- **at no loss of sensitivity**

i.e. performance characteristics not demonstrated in the Guggenheim paper or any other publication.

*Guggenheim et al. conducted their main analysis of the bandwidth and cavity length relation **using plano–concave micro resonators, not fiber-tip devices**, which they fabricated by controlled **deposition of droplets on glass substrates**. These results provide important insight into the relationship between cavity length and acoustic*

bandwidth and indicate that **a shorter cavity decreases the sensitivity** (a 250 μm cavity gives a NEP of 4 Pa, whereby a 30 μm cavity gives a NEP of 64 Pa).

Guggenheim et al. also fabricated two fiber-tip detectors and demonstrated how the bandwidth differs with cavity lengths. However, these results demonstrate that a shorter cavity of 16 μm increases the sensitivity to a NEP of 9.3 Pa, over 64 Pa for the 30 μm plano-concave cavity. This observation means that the analysis done on the plano-concave detectors does not directly translate to fiber-tip detectors and that **the results are not generalizable from plano-concave detectors to fiber-tip detectors**. Unfortunately, the sensitivity for the second 12 μm fiber tip detector is not reported. Therefore, **the Guggenheim paper also does not allow an insight on the relation of sensitivity to decreasing cavity size on fiber-tips**.

Now comparing the fiber-tip detectors shown in the Guggenheim paper, only their 16 μm fiber-tip detector reports on the sensitivity achieved, showing no loss of sensitivity but at a much lower bandwidth of ~ 25 MHz (at -6 dB) and lack of bandwidth homogeneity, as in Fig. A1. Neither the sensitivity, nor the frequency response of the second 12 μm fiber-tip detector is reported in the paper. Moreover, **while a 55 MHz response is reported, it is not indicated that this is the -6 dB point**, as commonly reported in ultrasound detectors. In fact, since the bandwidth of the 25 MHz @ -6 dB detector is reported as “40 MHz”, we assume that “55 MHz” is probably ~ 35 MHz @ -6 dB of a non-smooth distribution. These Guggenheim results are both $\sim 600\%$ lesser in performance to our design and not generalizable to our design, since ultra-small droplet formation dynamics, ultrasound edge diffraction, and beam propagation which affects the finesse all differ between the Guggenheim design and our design.

Nevertheless, if the reviewer can provide the frequency response and sensitivity of the 12 μm fiber-tip detector, we will include this report in the table below.

Figure A1: (a) Frequency response of the 16 μm fiber-tip device by Guggenheim et al. with a bandwidth of ~ 25 MHz

(b) Frequency response of the 6 μm fiber-tip new-design device from our work with a bandwidth of 150 MHz

Indeed, compared to the Guggenheim paper detectors, our detector uses a new tapered design which is shown to achieve the marked performance advance. This design is non-foreseen in the Guggenheim paper and uses controlled miniaturization of the cavity geometrical characteristics by building the cavity on a taper. In particular, what is not obvious in the Guggenheim paper is the particular relation of the bandwidth achieved, the uniformity, the SAW suppression and the sensitivity to the fiber design.

*Such marked performance advances cannot be seen as “incremental”. As mentioned also to the comments to reviewer #3, advances of this magnitude have historically been recognized as significant in biomedical imaging. **Optimization** typically means a process that better aligns, calibrates, tunes etc of a system, to **improve performance by 10-20%**. A new design leading to 300% improvement of optical detectors (and up to **600% over the fiber-tip detector that reports on sensitivity in the Guggenheim paper**), is by all definitions a fundamental advance, not an optimization.*

Also repeating from comments to reviewer #3, a general notion of improvement prediction and the actual demonstration of markedly improved performance using a different design are not obvious or predictable in the Guggenheim paper. Such improvements are the basis of scientific advance. If they were not, then miniaturization in the fields of micro-fluidics, nanotechnology etc, would also be a “predictable” reduction in fluidics or micro-particles. Or the Nobel prize awarded STED and PALM techniques in microscopy, would be “optimizations” of 4Pi microscopy, SNOM or structured illumination microscopy, that were already breaking the diffraction limit, before STED and PALM. However, prediction and experimental demonstration are not similar: Einstein predicted gravitational waves, but it took 60 years and another Nobel prize to experimentally demonstrate them by Hulse and Taylor. Without claiming that our work is anywhere close to these colossal science advancements, we use these examples to indicate that “obvious” cannot be validly claimed in retrospect. Else, one could also claim that PALM for example, “obviously” used localization microscopy and photoactivatable molecules, known decades before, to achieve a similar improvement in resolution, over previous techniques, as claimed by our detector.

To ensure accurate representation of the prior work by Guggenheim et al., we have updated the text and the comparison table to include the device with a cavity length of 12 μm and a 55 MHz bandwidth (presumably at -6 dB), noting that NEPD data was not provided for this device.

“Decreasing the cavity thickness alone leads to an increased radius of curvature, thereby increasing optical losses and reducing optical back-coupling efficiency into the fiber core. This trade-off was demonstrated on plano-concave micro-resonators on glass substrates for cavity lengths down to 30 μm [25], whereby a reduction in cavity size from 250 μm to 30 μm , increased the bandwidth but reduced the sensitivity. Conversely, when transferring this premise to the tip of optical fibers, the sensitivity achieved for a 16 μm cavity increased over the 30 μm plano-concave cavity, whereby the bandwidth decreased, indicating that the performance observed in plano-concave resonators is not transferable to fiber-tip resonators. This limited bandwidth and the omission of SAW suppression led to reduced lateral and axial resolution.”

...

Here, we introduce a paradigm shift in design, eliminating the need to offer a trade-off of the operational parameters,..."

	Dimensions		Bandwidth		Sensitivity		Resolution	
	Footprint	Thickness [μm]	-3 dB [MHz]	-6 dB [MHz]	NEPD [mPa.Hz ^{-1/2}]	NEPD x Area [mPa.mm ² Hz ^{-1/2}]	Lateral [μm]	Axial [μm]
TOF-PR	\emptyset 24 μm	6	110	149	1.5 (25 MHz)	9.0×10^{-4}	17	7
[24]	\emptyset 125 μm	21	25	30	11 (25 MHz)	1.3×10^{-1}	> 38*	> 35*
[25] ^Δ	\emptyset 125 μm	16	20	25	2.1 (20 MHz)	2.5×10^{-2}	> 47*	> 45*
[26]	\emptyset 125 μm	20	-	30	40 (30 MHz)	4.9×10^{-1}	84	231
[23]	\emptyset 125 μm	30	25	-	70 (25 MHz)	4.3×10^2	> 37*	>36*
[16]	3000 μm x 800 μm	9	-	230	9 (25 MHz)	9.9×10^{-7}	16	5

Table 1. Comparison of our detector (TOF-PR) with other optical fiber-based polymer resonators and SWEDs in the literature by dimension, bandwidth, and sensitivity. In the NEPD column, the frequency bandwidth used for characterizing NEPD is indicated in brackets. *Estimates for lateral and axial acoustic resolution are derived from the corresponding read-out beam size of the detector and its high cut-off frequency. In practice, the maximum achievable resolution will exceed the estimated values, primarily due to the coupling of surface acoustic waves to the effective detection region, which results in a detector aperture larger than the read-out beam size. ^Δ Only for the 16 μm detector all key parameters are available and reported in the table; for the 12- μm detector only a bandwidth of 55 MHz (presumably -6 dB) is given. Abbreviations: NEPD: Noise-equivalent pressure density, TOF-PR: tapered optical fiber-polymer resonator.

2) The claimed novelty reduces to a fabrication variant already implied by 2017

What the new devices clearly benefit from is smaller footprint due to tapering, which naturally enables thinner cavities and thus higher bandwidth—a trade-off that Guggenheim (2017) already analysed in depth (relationships among thickness L, footprint/base diameter, ROC, bandwidth/NEP). Without further proof, the present work reads as a miniaturized replication via a different fabrication route, rather than a substantive conceptual advance.

We thank the reviewer for this important concern as it has allowed to better explain the foundational novelty in our paper.

*A first foundational advance is that **the trade-off reported in the Guggenheim paper** does not apply to our design. Bandwidth, response homogeneity and SAW minimization all improved without loss of sensitivity!*

Second, as per answer to the previous comment, the results from Guggenheim are not generalizable, since they offer contradictory performance in the relation to the cavity dimensions used when moving from plano-concave to fiber-tip detectors.

Third the performance achieved by the Guggenheim fiber-tip detectors is 600% worse in bandwidth and exhibits a strongly non-uniform response over the detector in our work, as explained in detail in the response to the previous comment.

We further note that the comment of miniaturization being not a conceptual advance may not be justified. If it were true, and a **600% performance improvement over the Guggenheim on-fiber-tip detector, using a new design, is a “miniaturized replication”, then so is the development of MEMS, nano-technologies, lab-on-a-chip, microfluidics and semiconductor chips.** Likewise, our 2020 Nature SWED detector is mistakenly published, since it is “only” a miniaturized interferometer. And to extend this line of thought, so unjustified are the publications in Nature journals of a multitude of fluorescent protein variants shifting the wavelength in the past, or different designs of electrochemical sensors, all using electrochemical detection.

In contrast, as explained, the new design and unprecedented performance presented herein show a value and impact for on-fiber-tip detectors never before demonstrated, certainly not with the design in the Guggenheim paper, with all due respect to the importance of the work demonstrated therein.

We further note that although tapering can reduce the footprint of a cavity, it does not automatically lead to thinner cavities or improved imaging performance. It is possible to use a standard 125 μm single-mode fiber (SMF) to form a cavity length of 6 μm . However, in fiber-based devices, careful tuning of the cavity geometry is essential to increase back-coupling efficiency by reducing beam walk-off and achieving a uniform and broadband frequency response. This critical aspect was not considered in the on-fiber tip design described by Guggenheim et al. (2017).

The TOF-PR design features distinct cavity geometries for each cavity length to maximize optical phase sensitivity which increases the sensitivity of the detector and is demonstrated by the top-in-class NEPD of the 6 μm device (PR-III).

Additionally, the TOF-PR exhibits a flattened frequency response and reduces the impact of SAWs, which can introduce artefacts during image reconstruction. The wrong assumption that a high bandwidth automatically improves imaging resolution is incorrect. In fact, SAW interference can reduce resolution even when detectors have high bandwidth and a uniform frequency response (Shnaiderman et al. 2020).

Therefore, the primary advancement of TOF-PR lies not merely in increased bandwidth resulting from a smaller cavity length, but rather in the conceptual advance of a new design that optimizes the cavity geometry, yields a uniform bandwidth distribution, an unprecedented 300% - 600% improvement in bandwidth over the Guggenheim paper, minimization of the SAW interference and therefore fundamental advances in both axial and lateral resolutions, all achieved without a trade-off, as it does not compromise the sensitivity achieved, while enabling this unprecedented optoacoustic imaging resolution.

3) No parametric evidence for “independent” ROC–L control

The manuscript does not provide the necessary parametric, statistical demonstrations that ROC can be tuned at fixed L (and vice versa), nor that such decoupling yields performance beyond what “thinner L → wider bandwidth” already explains. In particular, the following are missing:

Controlled sets with fixed L, varying ROC (≥ 3 levels, $n \geq 5$ devices per level), and fixed ROC, varying L (≥ 3 levels, $n \geq 5$)—first on untapered 125 μm SMF, then on the tapered platform. Quantified L (from FSR), ROC (profilometry/white-light interferometry), footprint, finesse/Q, $-3/-6$ dB bandwidth, NEPD and NEPD \times Area, with statistics and uncertainty. Analysis that isolates contributions, e.g. $\partial\text{BW}/\partial L|_{\text{ROC}}$ and $\partial\text{BW}/\partial\text{ROC}|_L$, showing genuine independence rather than a single-parameter (L) effect.

We are afraid that we do not understand the comment, which reads mostly as a shorthand note rather than a structured argument. There is hardly a verb in the second paragraph.

Assuming that the comment is about evaluating the effects of the fiber’s curvature separately from the cavity length, we note that this comment is counter-intuitive from the previous arguments whereby all performance is apparently predicted by the Guggenheim paper.

Nevertheless, to support the claim that ROC and cavity length L can be independently selected, we evaluate the mode-match factor β of the plano-concave cavity (Bick et al., 2015; Klein, 2024). This factor quantifies how well the optical fiber mode matches the cavity mode, making it an effective metric for evaluating optical confinement within the resonator. For this analysis, the cavity was assumed to be spherical following Guggenheim et al.

The mode match factor β equals 1 when the mode field radius w_0 of the fiber is equal to the cavity mode radius w_c and is described by equation (1).

$$\beta = \frac{4}{\left(\frac{w_0}{w_c} + \frac{w_c}{w_0}\right)^2} \quad (1)$$

The cavity mode radius is given by:

$$w_c(L, \text{ROC}) = \sqrt{\frac{\text{ROC}\lambda}{\pi} \left[\frac{L}{\text{ROC}} \left(1 - \frac{L}{\text{ROC}} \right) \right]^{\frac{1}{4}}} \quad (2)$$

Using Equations (1) and (2), we show that the experimentally obtained (L, ROC) pairs for all three polymer resonators lie close to the theoretical maxima of β (see Fig. A2) (Klein, 2024). The fiber used in this work (PM1550-XP, Thorlabs) has a $1/e^2$ mode-field radius of $5.05 \pm 0.2 \mu\text{m}$ at a wavelength of 1550 nm. The same wavelength was used

for determining the cavity mode radius. This demonstrates that the mirror curvature is well matched to each corresponding cavity length, and that the fabricated devices operate close to the optimal mode-matching condition within the limits set by acoustic constraints. These constraints require keeping the cavity lengths small to ensure a broad bandwidth, reduced edge diffraction, and a uniform frequency response.

Figure A2: For each polymer resonator, the radius of curvature (ROC) is measured, and the mode match factor is computed according to equation (1). For each ROC, the maximum mode match is indicated with a star. Dashed lines show cavity lengths of the manufactured polymer resonators, which are positioned near the maximum mode match value corresponding to each ROC.

In addition, we gained valuable insights by performing a COMSOL Multiphysics (COMSOL AB, Stockholm, Sweden) simulation that compared light propagation in TOF-PR resonators to a planar fiber based Fabry-Pérot interferometer of equal cavity length, as shown in Figure A3. The simulation uses the Drude-Lorentz model to describe absorption in the silver mirrors but not scattering from grain boundaries or surface roughness. Nevertheless, the simulation accurately reproduces the experimentally observed interferometer transfer functions. The curvature of the cavity concentrates the E-field near the optical axis for PR-II and PR-III, whereas the planar Fabry-Pérot interferometer allows the E-field to leak beyond the mode field diameter due to beam walk-off. The lateral beam walk-off accumulates with number of round trips which increases the negative impact of the beam walk-off for high Finesse Fabry-Pérot interferometer. The beam walk-off is also evident from the reduced E-field intensity in the cavity for the planar detector compared to the cavities of the PR-II and PR-III detectors. The curved cavity design of PR-II and PR-III effectively reduces the sensitive area below the fiber's mode field diameter, which reduces the detector's sensitive area (aperture) and therefore increases its lateral resolution. Notably, PR-III has the smallest sensitive area, which correlates with its highest lateral resolution among the three detectors.

Figure A3: The E-field at resonance is shown for PR-II, PR-III, and a planar Fabry-Pérot interferometer with a cavity length of 6 μm . Both TOF-PR detectors focus the E-field, thereby minimizing the sensitive area (indicated by red line), whereas the planar detector demonstrates beam walk-off as indicated by a broadly distributed E-field with reduced intensity.

A like-for-like comparison against the Guggenheim fibre-tip devices under matched optical interrogation/reconstruction, to demonstrate surpassing performance not attributable merely to reduced L .

We thank the reviewer for suggesting a like-for-like comparison to further highlight the advantages of the TOF-PR design. In this study, image acquisition was performed by simultaneously scanning of excitation and detection, which enabled the resolution of blood vessels as small as 17.4 μm using wide-field illumination at a low fluence of 2 mJ/cm^2 .

For each detector, we provide B-scans and the corresponding PSFs, which clearly demonstrate the impact of bandwidth and SAW reduction on image resolution, PSF distortion, and artifacts. In comparison, even detectors with higher bandwidth, such as the SWED from Shnaiderman et al., exhibited significant PSF artifacts resulting from SAW interference, which limited the achievable minimum imaging resolution.

The TOF-PR detector provides a more uniform frequency response than the fiber-tip device of Guggenheim et al. Simply reducing the cavity length of a standard single mode fiber is not enough to match the TOF-PR detector's performance. Otherwise, Guggenheim et al. would have achieved similar imaging resolution with their device.

In contrast, the system described by Guggenheim et al. used a stationary detector in transmission mode with a scanned, focused excitation beam with a fluence of 2100 mJ/cm². This configuration corresponds to optoacoustic microscopy and is inherently restricted to superficial imaging confined to the optical focus (Omar et al. (2019)). Consequently, both the lateral (R_L) and axial (R_A) resolutions are limited by the low cut-off frequency ($f_{cut-off}$) and the detector aperture (Φ_d), as described by equations (3) and (4) following Shnaiderman et al. (2020). Consistent with these equations, an axial resolution of 36 μm was reported.

By contrast, we optimized the TOF-PR detector for a mesoscopic implementation under diffusive illumination. As such, a like-for-like comparison is impractical due to the large difference in fluence and imaging modality. Reproducing the experiments on the exact imaging setup described by Guggenheim et al. would demand substantial extra effort and would suggest that every fiber optic ultrasound detector must be evaluated using the same imaging modality as Guggenheim.

$$R_A \approx 0.8 * \frac{v_s}{f_{cut\ off}} \quad (3)$$

$$R_L = [R_A^2 + \Phi_d^2]^{\frac{1}{2}} \quad (4)$$

Major revision:

Require (i) correction of the prior-art statements and Table; (ii) the parametric, statistically powered ROC–L independence on both untapered and tapered platforms as outlined; and (iii) a matched-condition comparison to Guggenheim’s fibre-tip devices.

If the authors cannot provide these data, the current contribution appears to be an incremental fabrication variant whose novelty likely falls short of the journal’s threshold.

The response to this statement, and the reasons that it is unjustified, has been explained above in detail.

Literature

Bick, A., et al. (2016). "The role of mode match in fiber cavities." *The Review of scientific instruments* 87 1: 013102.

Klein, F. (2024). Experiments towards strong coupling and ground state cooling in an atom-optomechanical hybrid system. *Physics*. Hamburg, University Hamburg. PhD Thesis